

# Non-linear aspects of the tidal dynamics in the Sylt-Rømø Bight, south-eastern North Sea

Vera Fofonova[1,2], Alexey Androsov[1,3], Lasse Sander[2], Ivan Kuznetsov[1], Felipe Amorim[2], H. Christian Hass[2], Karen H. Wiltshire[2]

[1] Alfred Wegener Institute, Helmholtz Centre for Polar and Marine Research, Bremerhaven, 27570, Germany
[2] Alfred Wegener Institute, Helmholtz Centre for Polar and Marine Research, List/Sylt, 25992, Germany
[3] Shirshov Institute of Oceanology, Moscow, 117997, Russia

*Correspondence to*: Vera Fofonova (vera.fofonova@awi.de)

**Abstract.** This study investigates the tidal dynamics in the Sylt-Rømø bight with a focus on the non-linear processes. The FESOM-C model was used as the numerical tool, which works with triangular, rectangular or mixed meshes and is equipped with a wetting/drying option. As the model's success at resolving currents largely depends on the quality of the bathymetric data, we have created a new bathymetric map for an area based on recent studies of Lister Deep, Lister Ley, and the Højer and Rømø Deep areas. This new bathymetric product made it feasible to work with high resolution grids (up to 2 m in the wetting/drying zone). As a result, we were able to study the tidal energy transformation and the role of higher harmonics in the domain in detail. For the first time, the tidal ellipses, maximum tidally-induced velocities, energy fluxes and residual circulation maps were constructed and analysed for the entire bight. Additionally, tidal asymmetry maps were introduced and constructed. The full analysis was performed on two grids with different structures and showed a convergence of the obtained results as well as fulfillment of the energy balance. A great deal of attention has been paid to the selection of open boundary conditions, model verification against available tide gauges, and recent ADCP data. The obtained results are necessary and useful benchmarks for further studies in the area, including baroclinic and sediment dynamics tasks.

## 1 Introduction

The Sylt-Rømø Bight (SRB) is one of the largest tidal catchments in the Wadden Sea, which stretches from the Dutch island of Texel to Skallingen, a peninsula in Denmark. The SRB is characterized by the barrier islands Sylt (Germany) and Rømø (Denmark), which are separated by a tidal inlet called Lister Deep. Beginning in the last century, two artificial causeways, the Hindenburg Damm (1927) and the Rømøvej (1948), have connected the islands Sylt and Rømø to the mainland and thus created a semi-enclosed back barrier environment. Water exchange with the North Sea takes place through the 2.8 km-wide Lister Deep. The main channels draining the tidal back barrier environment are called Lister Ley, Højer Deep and Rømø Deep (Fig. 1).

The bight is characterized by large intertidal areas occupying about 40 % of the entire bight. The considered domain has an average water depth of ~4 m and a maximum water depth of ~37 m in Lister Deep (Fig. 1). The tidal range in the area is ~1.8



m (e.g., Pejrup et al., 1997) and the water column is generally well mixed (e.g., Villarreal et al., 2005). The annual mean freshwater discharge into the bay is only ~7 m$^3$/sec (e.g., Purkiani et al., 2015).

The tides play a major role in the local bight dynamics. Estimates of maximum horizontal tidal velocities in Lister Deep vary
from 1.2 to 2 m/s. Estimates of the mean water volume entering the basin during flood and leaving during ebb (the tidal prism) vary from 4 to 6.3 *10$^8$ m$^3$ (Bolaños-Sanchez et al., 2005; Gätje and Reise, 1998; Gräwe et al. 2016; Lumborg and Windelin, 2003; Nortier, 2004; Pejrup et al., 1997; Purkiani et al. 2015). The residual circulation of the M2 wave, the main tidal constituent in the area, is presented in  Burchard et al. (2008)  and Ruiz-Villarreal et al. (2005)  (wherein the grid resolution is 200 m); it shows maximum values up to 0.3 m/s in the area of Lister Deep, Lister Ley and Højer Deep edges
(the zones of large bathymetric gradients). There are two large vortex structures located at the entrance of the Königshafen embayment and directly north of the island of Sylt in the western Lister Deep (Fig. 1). They have clockwise and anticlockwise directions of rotation, respectively.

There is a pronounced asymmetry in the tidal water level and current velocities behaviour, caused by complex morphological features and by the general shallowness of the research area (e.g., Austen 1994; Becherer et al., 2011; Lumborg and
Windelin, 2003; Nortier, 2004; Ruiz-Villarreal et al., 2005). It is known that the Lister Deep can be characterized as an ebb-dominated area, which means that the velocities during ebb are larger compared to flood velocities in the mean and maximum senses (e.g., Hayes, 1980; Oost et al., 2017; Fig. 1).  The analysis of bedforms based on seismic profiles revealed that the area around Lister Deep is represented by a complex spatial pattern of the flood- and ebb-dominated subaqueous dunes (Boldreel et al., 2010).

The tidal residual circulation and asymmetric tidal cycles largely define the transport and accumulation of sediment and the distribution of bedforms in the bight (e.g., Boldreel et al., 2010; Burchard et al., 2008; Hayes, 1980; Nortier, 2004; Postma, 1967). Note that the tide entering the Wadden Sea leaves ~3.5 * 10$^6$ t yr$^{-1}$ of suspended matter there according to Postma (1981). The most intensively studied dynamic is in the area of Lister Deep (a 'bottleneck' area); it is studied more than the other subareas as it can shed light on the sediment as well as water and salt budgets of the whole bight (e.g., Kappenberg et
al., 1996; Nortier, 2004; Bolaños-Sanchez et al., 2005; Becherer 2011; Purkiani et al. 2016; Gräwe et al. 2016; Lumborg and Windelin, 2003; Lumborg and Pejrup, 2005; Burchard et al., 2008; Purkiani et al., 2015; Villarreal et al., 2005). However, the listed studies offer quite different estimates of suspended matter fluxes and the sediment budget. In our opinion, one of the reasons for such disagreement is a gap in understanding of the role of the tide. In a back-barrier environment like this, higher tidal harmonics take on a relatively large role in the dynamics. For example, Stanev et al. (2015) demonstrated that a
higher harmonic (M4) in the German Bight causes strong tidal asymmetry. Though continuous observational data can give answers about tidal water level and velocity behaviour at the current position, it is nearly impossible to extrapolate this information to a larger area due to strong non-linear processes. The dynamic impact of higher harmonics is not always reproduced by the numerical model due to both coarse grid resolution and numerical limitations. Among these, the missing wetting/drying option plays a major role (e.g., Stanev et al., 2016). It is also clear that, in a region such as the SRB, the





quality of the bathymetry and the resolution of its features are the key points for realistic model calculations of the hydrodynamics and sediment transport. To our knowledge, the best resolution among the available numerical simulations of the area is 100 m (Purkiani et al., 2015).

This study investigates tidally induced dynamics in the SRB based on the FESOM-C coastal numerical solution (Androsov et al., 2019) with a focus on the non-linear dynamics. The FESOM-C model works with triangular, rectangular and mixed
meshes and is equipped with a wetting/drying option. This allowed us to resolve the dynamics in the intertidal zone carefully. There have been three studies of the Lister Deep, the Lister Ley, and the Højer and Rømø Deeps in recent years (Mielck et al., 2012, Boldreel et al., 2010). Based on these, we have created a new bathymetric map for the area. For this new bathymetric product, it made sense to work with high resolution grids (up to 2 m in the wetting/drying zone). We have studied the evolution of the tidal energy in the domain in detail by providing residual circulation and tidal ellipse maps,
defining the role of higher harmonics in the dynamics and by suggested and realized tidal-wave asymmetry analysis. We have not only concentrated our attention on M2 wave-induced dynamics but have also prescribed the elevation generated by the sum of M2, S2, N2, K2, K1, O1, P1, Q1 and M4 harmonics at the open boundary. The main motivation is that M2 accounts for only 60 % of the tidal potential energy entering the domain based on available tidal atlases (e.g., TPXO 9). A great deal of attention has been paid to the choice of open boundary conditions, model verification against available tide
gauges as well as against recent ADCP data, and behavior of the solution on different grids. The last item is important because, if we are sure the results converge, then further increasing the resolution will not lead to substantially different results. We also showed the intertidal zone and updated the tidal prism value. This study is a necessary and useful benchmark for the further studies of baroclinic and sediment dynamics in the area.

The manuscript is organized as follows. The 'Model Setup' section contains information about the numerical ocean
solution, the grids, the open boundary conditions and the bathymetric data we used. The next section contains information about the observational data we used to verify the simulations. The 'Results' section contains model validation results and detailed information about tidally induced barotropic dynamics in the area, with a focus on the nonlinear dynamics. The 'Discussion' section contains information about possible sediment dynamics outputs based on our results, and about grid performance. The last section summarizes the manuscript.

## 2 Model setup

### 2.1 Coastal numerical solution
FESOM-C is a coastal branch of the global Finite volumE Sea Ice Ocean Model (FESOM2, Danilov et al, 2017). FESOM-C has cell-vertex finite volume discretization and works on any configurations of triangular, quadrangular or hybrid meshes (Androsov et al., 2019; Danilov and Androsov, 2015). It has split barotropic and baroclinic modes and a terrain-following
vertical coordinate; and it is equipped with 3rd-order upwind horizontal advection schemes, implicit 3rd-order vertical

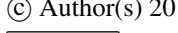



advection schemes, implicit vertical viscosity, biharmonic horizontal viscosity augmented to the Smagorinsky viscosity, and the General Ocean Turbulence Model (GOTM, Umlauf and Burchard, 2005) for the vertical mixing. The wetting and drying of intertidal flats have been included because this is a crucial point for the reconstruction of the non-linear dynamics in the shallow zone.

All results demonstrated below (except inter-comparison of the different tidal solutions) are obtained based on multi-layer barotropic simulation with tidal forcing only. The 10-sigma vertical layers (with crowding near the bottom) were prescribed, and the roughness-height was set to 0.001 m. The turbulence closure was represented by second-order model with coefficients from Canuto et al. (2001) (model A, realized in GOTM library). Unless otherwise indicated, the pictures visualize depth-averaged behavior.

**2.2 Grids**

We have created two grids for the area of interest; all simulations were performed on these two grids. One grid is curvilinear and referred to below as the "first grid." The other is unstructured and contains mostly arbitrary quads with a few triangles; below it will be called the "second grid."

The first grid contains 119305 nodes; the resolution varies from 14 to 261 m. The finest resolution is in Lister Deep near the 110 southwestern boundary and in the eastern area of the internal part of the domain. The first grid was generated by the elliptic method (Thompson, 1982).

The second grid contains 208345 nodes (10398 triangles; 201141 quads); its resolution varies from 2 m in the wetting/drying zones to 304 m in the deepest area of the external (seaward) part of the considered domain (Fig. 1). The matrix of element sizes is based on the information about the bathymetry, the bathymetry gradient and the zones of particular interest (Lister 115 Deep; main draining channels). The second grid was built using the mesh generation software package of the Surface Water Modeling System (SMS version 12.3, AQUAVEO).

**2.3 Open boundary conditions**

We relied on four sources for the open boundary conditions of the tidal elevation: TPXO 8.1 and 9 (Egbert et al., 2002); the output of NEMO (Nucleus for European Modelling of the Ocean, Gurvan et al., 2017) simulations for the North-West 120 European Shelf; and FES2014 (Finite Element Solution 2014, Carrere et al., 2016).

TPXO 8.1 and 9 are fully-global models of ocean tides which best-fit, in a least-squares sense, the Laplace Tidal Equations and altimetry data. They provide information about 13 harmonic constituents (or 15 with TPXO 9). TPXO 8.1 and 9 atlases are combinations of the 1/6 degree base global solution and the 1/30 resolution local solutions for all coastal areas including our domain of interest. Each subsequent model in TPXO is based on updated bathymetry and assimilates more data than 125 previous versions.





The NEMO product includes information about hourly instantaneous sea level. The sources are full baroclinic simulations based on version 3.6 of NEMO with data assimilation (vertical profiles of temperature, salinity, and satellite sea-level anomaly). The model is forced by lateral boundary conditions from the UK Met Office North Atlantic ocean forecast model and, at the Baltic boundary, by the CMEMS Baltic forecast product. Information about the tidal constituents we obtained by
performing an FFT (Fast Fourier Transform) analysis of the elevation signal at our open boundary based on data for one year (2017).

FES2014 is a global finite-element hydrodynamic solution with assimilated altimeter data and a grid resolution of 1/16° x 1/16°. FES2014 is the latest version of the FES (Finite Element Solution) tide model; it was developed from 2014 to 2016 and provides a solution for the 34 main constituents.

**2.4 Bathymetric data**

Bathymetric data for a given area were generated based on two sources: the AufMod database (Valerius et al. 2013), with a 50 m resolution for the whole area; and newly obtained data for the inlet and the main tidal channels, with a grid resolution of 10 x 10 m. The high-resolution part was not modified; the transition zone between coarse and fine bathymetric data was smoothed by various convolution filters.

The new bathymetric information in Lister Deep tidal inlet and the main tidal channels was obtained during winter of 2017-2018 using the hull-mounted ELAC SeaBeam 1180 multibeam system on board the RV "Mya II" of the Alfred Wegener Institute. The data were collected at sub-meter resolution and resampled to a gapless 10 x 10 m grid for the purpose of this study. The multibeam survey was intended to cover areas with water depths of mainly > 5 m in the tidal inlet and the main tidal channels. Data on the depth of areas of shallow water (< 5 m) or that were otherwise inaccessible to the vessel were
obtained from the AufMod database (Valerius et al. 2013) and used with a spatial resolution of 50 x 50 m.

**3 Observational data for model validation**

**3.1 ADCP and wind data**

The observed velocities represented the base for verification of the model and testing of the different tidal open boundary conditions. The data set is composed of observed profiles of the water currents gathered on five cruises of the RV "Mya II".
The transects cross Lister Deep perpendicularly in two positions: for outer part, at the narrowest and bathymetrically simplest part of the inlet and for the inner part, because the entire water volume has to pass through this transect; and at the entrance areas to the three main tidal channels (Fig. 1). The cruises were conducted between May 22 and May 30, 2018; observations were carried out for half and whole tidal cycles (Table 1). The data were collected using a Teledyne RDI WorkHorse 600 kHz (Teledyne RD Instruments, San Diego, USA) mounted in the moon pool of the RV "Mya II" (of the
Alfred-Wegener-Institute) with an offset of 1.3 m related to the water surface. In addition, a differential GPS with a motion sensor worked together with the ADCP to refine and correct the velocity measurements regarding the heading, pitch and roll





movements of the ship. Here we would stress that the measurements were done during different tidal periods (Table 1), which is crucial for high-quality verification.

Wind data were automatically measured by an anemometer mounted on the RV "MYA II"; the data were obtained from the
DAVIS SHIP database for the time of the cruises in order to get an impression about the local wind conditions during the surveys. The main wind direction during the cruises was around 90° (east); the most frequent intensities were in the range of 5 to 10 m/s. On May 24 the wind was blowing strongly from the east for nearly the whole cruise. Exceptions concerning wind direction were the cruise on May 23, when winds were from the NNW (330°), and the cruise on May 29, when the most frequent intensity ranged between 10 and 15 m/s.

**3.2 Tide gauge (TG) data**

The tide gauge data for the area are represented by three stations: ListTG (8.441°E; 55.017°N), VidaTG (8.6667°E; 54.9668°N) and HavnebyTG (8.5654°E; 55.087°N) (Fig. 1). All data were downloaded from the EMODnet database (the European Marine Observation Data Network Seabed Habitats project (www.emodnet-seabedhabitats.eu)), which is funded by the European Commission's Directorate-General for Maritime Affairs and Fisheries (DG MARE) and provided by the
Waterways and Shipping Office in Tönning, the Danish Meteorological Institute and the Danish Coastal Authority. For the spectral analysis, we used data with a 10-minute resolution covering the time period from the middle of 2014 through the end of 2017. The time of the observations is the UTC (+0) zone.

**4 Results**

**4.1. Model validation**

Model validation was organized into the following two steps: (1) selection of the best open boundary conditions using available ADCP data and (2) validation of the best open boundary solution against existing tide gauge data, in particular using List TG, Vidå TG and Havneby TG data. For step (1), the measurements were done during different tidal periods (spring, neap, ebb, and flood) in the area of Lister Deep and the main inlets (Fig. 1), which are characterized by the largest depths (up to 35 meters) in the domain and by the highest as well as by complex tidally induced velocities. We performed
the frequency analysis using the MATLAB-package T-TIDE (Pawlowicz et al., 2002) and identified the errors in amplitude and phase for the main tidal constituents (with maximum amplitudes), including higher harmonics.

Model validation was organized into the following two steps: (1) selection of the best open boundary conditions using available ADCP data;  (2) validation of the best open boundary solution against existing tide gauge data, in particular using List TG, Vidå TG and Havneby TG data. During the second step we performed the frequency analysis using MATLAB-
package T-TIDE (Pawlowicz et al., 2002) and identified the errors in amplitude and phase for the main (with maximum amplitudes) tidal constituents, including higher harmonics.



Table 2 represents the Root Mean Square Deviation (RMSD) and correlation coefficients of the observed velocities (ADCP data) and the modelled velocities based each open boundary condition. The comparison is based on depth-averaged velocities since the measurements were performed in the deep part of the domain and our task here was to check the performance of the different tidal forcing. In table 2, results for all the open boundary solutions we used are shown only for the first grid since the difference between the results on different grids is less than 0.01 for correlation coefficients and 0.01 m for the RMSD. However, we would note that the second grid provides slightly better results despite its somewhat coarser resolution (the second grid has a larger number of cells, mainly due to detailed representation of the wetting/drying zone). Based on additional experiments (not shown), we think the reason is that the second grid reflects the bathymetric gradients.

We used different bottom-friction coefficients ($C_d$) for the different tidal solutions. At the beginning, we took the $C_d$ coefficient as equal to 0.0025 for all simulations. The NEMO solution got much worse results in this case than presented in table 2 in terms of the RMSD and correlation coefficient. Further analysis showed that the predicted velocities for the NEMO open boundary solution are too large during some tidal phases. We therefore decided to vary the $C_d$ in a range from 0.002 to 0.004, taking into account the largely sandy bed in the domain, to reach the best agreement with observations (e.g., Werner et al., 2003). Finally, for the TPXO solutions, we used a $C_d$ equal to 0.0025; and for the NEMO solution, a $C_d$ equal to 0.0035. Note that smaller or larger coefficients lead to the same or worse results in terms of RMSD and the correlation coefficient.

Table 2 shows that the TPXO 9 solution fits the ADCP data best; second best are the TPXO 8.1 and NEMO solutions; and then the FES 2014 solution follows. For all solutions except FES, the correlation coefficients are higher during spring tides as well as in the deepest part of the domain; this is true in particular for the measurements performed on May 29 and May 30 and despite the quite strong winds often ranging from 10 to 15 m/s (Table 1, Fig. 1). On May 23 and May 24, the measurements were performed on nearly the same side; but the May 24 correlation coefficient for the 'v' component is relatively small. This can probably be explained by the permanent wind from the east. We can conclude that tides in that zone explain, on average, more than 80 % (or 90 % or more in case of a spring tide) of the dynamics in case of absent storm (more than 20 m/s) and blowing continuously in one direction winds. Likewise we conclude that it would be impossible to judge open-boundary-condition quality using information from only one cruise. Figure 2 represents observed and modelled depth-averaged velocities on May 29 (a spring tide). Figure 2 clearly illustrates how the different solutions can align closely at one moment and then begin to deviate greatly at another. Furthermore, the largest velocity errors for all solutions occur when tidal velocities are small as well as during the tidal state change (see e.g., in Fig. 2, the time from 7 a.m. to 8 a.m. or from 1 p.m. to 2 p.m.). This is quite logical, because other effects such as baroclinicity, wind impact, and their non-linear interactions then become more pronounced. However, TPXO 9 shows the best agreement with observations during slack tide and provides the smallest RMSD among suggested solutions for all observation days (Fig. 2).

Despite the quite good results of TPXO 8.1, the second grid yields a number of vortex structures near the open boundary that cannot be removed by the sponge layer, which dumps advection and diffusion near the open boundary. It is known that grids





that employ an arbitrary normal at the open boundary are subject to the quality of the open boundary signal (Danilov and Androsov, 2015). For TPXO 8.1, the behaviour of the phase is not realistic near the solid boundary: the phase goes through zero simultaneously near the western and eastern solid boundaries.

Once we determined the best open boundary conditions solution—TPXO 9, based on ADCP data—we moved on to the second verification stage. For this, we switched to a 3D-simulation with 10 vertical layers, which are crowding near the bottom. The optimal roughness height was 0.001 m; this value agreed with one estimated from observations in a similar region (Werner et al., 2003) and, in terms of the mean, with a Cd equal to 0.0025 for the 2D-scenario. Table 3 presents the results of the frequency analysis of the observational data from the List, Havneby and Vidå TGs as well as from the model using TPXO 9 open boundary conditions. Here we present results from both of the grids under consideration. The observed phase values are quite often situated between the values given by the solution on the different grids. This is because we used nearest-to-the-observational-point grid nodes to perform the analysis, and did not interpolate the values to the particular coordinates. Either solution (per grid) performed well; but each slightly underestimated the amplitudes for major diurnal and semidiurnal constituents (except for the M2 amplitude at the Vidå TG). Note that the M2 tidal wave and its subharmonics are shown to vary in amplitude and phase on seasonal, annual, and secular time-scales (Müller, 2012, 2014; Woodworth et al., 2007; Gräwe et al., 2014). The annual variations of the M2 amplitude are between 6 and 11 % of the annual mean amplitude for the shallow water stations; the phase can vary by 2 to 6º persistently over the last century (Gräwe et al., 2014). The summer amplitudes seem to be larger than the winter amplitudes; this can be explained by changes in thermal stratification (Müller, 2012, 2014; Gräwe et al., 2014). The same probably holds for all major semidiurnal and diurnal components. Our simulations obtain "winter" amplitudes, while the observational data provide a mean annual amplitude. The M4 constituent also has a noticeable phase error of ~30° at all stations. But due to large bathymetric uncertainties in the area of the stations, such an inaccuracy is hard to correct (here, the resolution of the bathymetry we used is 50 m). Note that at the Vidå TG, the second grid shows significantly better agreement than the first grid. This is not surprising since the second grid has much better resolution there.

### 4.2 Wetting/drying, maximum tidally induced velocities and tidal prism
Figure 3a shows the probability, in the second grid case, of each node being wet with tidal forcing alone. This figure is based on simulation results for two full tidal cycles (29.5 days*2) after the model reaches a periodic regime. The second grid more accurately represents the wetting/drying zone; it can be traced in smoother forms and in smaller-scale detail than in the first grid (not shown).

An extensive intertidal subarea situated in the western and southern parts of the considered domain as well as in the in the Königshafen embayment can be seen; and the Jordsand creates a secondary bight with the Rømø Deep main channel (Fig. 1, Fig. 3a).





Figure 3b shows the maximum velocities at each grid point within a full tidal cycle (29.5 days). Thus, Fig. 3b shows the highest possible tidally induced velocities in the domain. Figure 3b exhibits, as expected, a correlation with the depth (Fig. 1); but there are many peculiarities, which emphasize the large role of non-linear processes in the domain. The maximum velocities can be found at the opening of Lister Deep and near the edge of Sylt during spring ebb and are ~1.98 m/s. The tidal prism for the bight varies from 3.3 to 6.5 $*10^8$ m, with a mean value of 4.8 $*10^8$ m; the tidal prism for the whole area varies from 5 to 10 $*10^8$ m, with a mean value of 7.5 $*10^8$ m.

### 4.2 Energy balance

The analysis of the energy budget and energy flux distribution provides an important insight into the evolution of energy in the modelled region. The energy balance for the vertically averaged equations for the barotropic case has the following form (see, e.g., Androsov et al., 2002):

$$\frac{\partial \overline{E}}{\partial t} + \nabla \cdot \left[ \rho H \left( g\xi + \frac{1}{2}|\overline{\mathbf{v}}|^2 \right) \overline{\mathbf{v}} \right] = -\rho C_d |\overline{\mathbf{v}}|^3 + \rho \overline{\mathbf{v}} \cdot \left( \nabla \cdot (KH\nabla \overline{\mathbf{v}}) \right),$$ (1)

where $\overline{E} = \frac{1}{2}\rho(H|\overline{\mathbf{v}}|^2 + g\xi^2)$ is the total energy per unit area, $\overline{\mathbf{v}} = \int_{-h}^{\xi} \mathbf{v}\, dz$ is the vertically integrated fluid velocity, $\mathbf{v} = (u, v)$, $\xi$ is the sea surface elevation, h is the water depth, $H = h + \xi$ is the full water depth, $\rho$ is the water density, $C_d$ is the bottom drag coefficient, K is the horizontal eddy viscosity coefficient, g is the acceleration due to gravity and $\nabla = \left( \frac{\partial}{\partial x}, \frac{\partial}{\partial y} \right)$ is the nabla operator. Note that we did not introduce horizontal viscosity into our equations; therefore $K$ equals 0 in our simulations, and further this part of the balance will be omitted. After integration of eq. (1) over the region $\Omega$ with boundary $\partial\Omega = \partial\Omega_1 + \partial\Omega_2$, where $\partial\Omega_1$ is the solid part of the boundary and $\partial\Omega_2$ is the open boundary, taking into account the Gauss formula for divergence and the condition of zero velocities at $\partial\Omega_1$, we obtained the following mean energy balance equation for our depth-averaged solution:

$$\int_{\Omega} \frac{\partial \overline{E}}{\partial t}\, dx\, dy = -\int_{\partial\Omega_2} \left[ \rho H \left( g\xi + \frac{1}{2}|\overline{\mathbf{v}}|^2 \right) \overline{\mathbf{v}} \cdot \mathbf{n} \right] ds - \int_{\Omega} \rho C_d |\overline{\mathbf{v}}|^3\, dx\, dy,$$ (2)

where $\mathbf{n}$ is the outward normal to $\partial\Omega_2$.

The first term on the right side of (2) is the total flux of energy across the open boundary, and the second term is the rate of energy dissipation due to the bottom friction. Figure 4 shows the energy balance for the whole area based on second grid for the summary tide for one M2 period (the energy balances for the whole area at both grids are visually identical). Once we have different diurnal and semidiurnal constituents in the system, the energy balance every M2 period will not be the same. However, Fig. 4 demonstrates the overall picture. The total energy in the area varies between $7*10^{11}$ to $3.5*10^{12}$ J on both grids (in the case of TPXO 8.1 conditions, the total energy in case of second grid had different variation limits). For





comparison, the full tidal energy in the modelled area is almost equivalent to the full energy of the barotropic tidal dynamics
in the Strait of Messina (Androsov et al., 2002), and one order of magnitude less than in Bab el Mandeb Strait (Voltzinger
and Androsov, 2008). The potential energy is one order of magnitude larger than the kinetic energy. The shares of the total
energy based on the open boundary information are distributed in the following way: M2 brings 58%, S2 - 13.5%, N2 - 10%,
K2 - 3.5%, K1 - 4%, O1 - 5.5%, P1 - 1%, and the Q1 and M4 components bring 1.5% and 3% correspondingly. The fluxes
through the open boundary can remove up to $1.5 *10^8$ J per second. Bottom friction takes up a significant amount of energy,
on average $6.4 *10^7$ J per second, due to the shallowness of the bight (Fig. 4). The imbalance is two orders of magnitude
smaller than the energy change. The second grid is a bit more dissipative; its imbalance is 10% larger than that on the first
grid. This is for several reasons: the second grid in particular contains triangles, arbitrary quads and, in some places, large
gradients in grid cell size, which causes additional noise (Danilov and Androsov, 2015). We would point out that it is hard to
estimate the numerical dissipation rate precisely. The current imbalance consists not only of numerical dissipation but also of
uncertainties in the energy balance calculation once we no longer accounted for the role of the explicit time-difference
scheme with an Adams–Bashforth extrapolation and the effects of velocity filtration. Our additional runs showed that the
impact of these procedures has the same magnitude as a calculated imbalance. We think that, on the first grid, the "real"
imbalance due to numerical viscosity is close to zero, at least one magnitude smaller than the presented one.

The maximum change in system energy and maximum fluxes through the open boundaries take place during ebb tide; the
ebb phase duration is, on average, 0.85 of the flood phase. These general conclusions mask very patchy dynamics in the area,
which are considered in detail in the next sections.

Figure 5 demonstrates the energy fluxes in the area. The tidal energy flux, represented by the sum of the potential and kinetic
energy fluxes, is estimated using the following definition (Crawford, 1984; Kowalik and Proshutinsky, 1993):

$$(E_\lambda, E_\theta) = \frac{1}{T}\int_0^T \rho H \left( g\xi + \frac{1}{2}|\bar{\mathbf{v}}|^2 \right) \bar{\mathbf{v}} \ dt, \tag{3}$$

where $E_\lambda, E_\theta$ are the zonal and meridional components of the tidal energy flux vector, and T is the full tidal period (sinodical
month - 29.5 days). Figure 5a, b, and c show that the potential energy fluxes in the area are one order of magnitude larger
than the kinetic energy fluxes. Therefore, total energy fluxes are largely defined by potential energy fluxes (Fig. 5b, c). It can
be noted that the directions of the potential and kinetic energy fluxes can deviate from each other and be opposite (Fig. 5a,
b); this is can be explained easily by the fact that the kinetic energy fluxes largely reflect dissipation of the energy, and the
potential energy fluxes largely reflect energy distribution pathways in the domain. So, in Fig. 5b and c, it can be seen that the
dynamics in the external part of the domain is determined by the Kelvin wave coming from southwest. Obviously, the bulk
of the energy comes into the internal part of the domain through Lister Deep. The energy leaves the whole modelling domain
mostly through the north-eastern part of the open boundary. Figure 5d demonstrates that the bulk of the energy comes to
Königshafen from the south and there circulates through the system of gyres (the curl part of the energy fluxes is larger than





the divergence part). Figure 5d also indicates that the energy fluxes along the coastline in this subarea are opposite in direction to those within the main channel.

## 4.3 Residual circulation

The residual circulation in the area is characterized by the large number of vortex structures with different rotation directions

in the area of Lister Deep and the main channels (Fig. 1, 6a). Note that the residual circulation in the external part of the domain is defined by kinetic energy fluxes, because here the role of nonlinearity in the continuity equation is minor (the water depth compared to the tidal amplitude is relatively large). These vortexes organize a complex residual circulation pattern; the strongest circulation (up to 0.45 m/s) can be found in the area of Lister Deep, where different incoming and outgoing flows meet, and in the other areas of large bathymetric gradients. Note, that the two largest vortex structures in

terms of the residual velocities magnitude, which show opposite rotational, were also resolved by Burchard et al. (2008) and Ruiz-Villarreal et al. (2005) based only on the M2 signal. Near Königshafen we can also follow the gyre system in the opposite direction, which alternates (Fig. 6b). Note that, here, the energy fluxes and residual circulation patterns have much in common. The bulk of the energy fluxes come from the south, and tides lose energy on the way, causing residual circulation shaped by the bathymetric features. For this subarea, the residual velocities are less than 0.05 m/s. The residual

circulation in the external part of the domain shows that the bulk of water masses penetrate to the internal part of the domain from the western boundary and the south, moving along the coast. Water masses mainly leave the domain through the central part of the open boundary. Such a complex residual circulation (Fig. 6) signals potential difficulties in calculating sediment budget.

## 4.4 Tidal ellipses

The resulting movement in the SRB represents a superposition of the tidal waves reflected off the region's solid boundaries. As a result of this interference, a standing wave (or seiche) occurs, containing only one component of the velocity for the main tidal harmonics, which is to say: currents are close to reverse. The Coriolis effect does not lead to significant cross-currents. Figures 7a and 7b show the ellipses of the wave M2 and S2 tidal currents, respectively; the character of the currents of either wave component is practically indistinguishable in the SRB except for the amplitude of the M2 velocity, which

turned out to be almost four times larger than the amplitude of the S2 velocity. The orientation of the tidal ellipse is determined by the sign of the ellipticity. A negative ellipticity value means that the tidal current vector rotates anticyclonically. The anticyclonic rotation of the depth-averaged velocities near Sylt-Rømø is inherent only in zones of a significant bathymetric gradient where maximum tidal currents are observed. This kind of rotation has to overcome the Coriolis effect, which spins in the opposite direction. At the same time, on the boundary with the North Sea (the open

boundary of the modelled region), the currents have a constant cyclonic rotation, which corresponds to the cyclonic movement of the Kelvin wave along the coast in the Northern Hemisphere.



The ellipses induced by the M4 wave are shown in Fig. 7c. This picture differs from the main tidal harmonic frequencies in the presence of mixed flow zones. In deep-water channels, the currents are close to reverse; and on the shape of bathymetry the transverse velocity component appears. The amplitude of the velocity of the M4 wave reaches 25 cm/s in most

bottlenecks of the modelled region and near Königshafen. Near the open boundary, the currents have a pronounced anticyclonic character, i.e., nonlinear flows overcome the Coriolis effect, spinning the flow in the opposite direction. In some areas around the shallow zones, the major axis of the M4 ellipsis is at a significant angle to the major axis of the M2 ellipsis. This can lead to secondary circulation in a rotating system.

### 4.5 Nonlinearity structure and tidal asymmetry

The major higher harmonics in the area are $M_4$ with 36.6% of total energy attributed to the nonlinear harmonics in the whole considered domain, $M_6$ - 12.43%, $MS_4$ - 12.27%, $MN_4$ - 10.95%, $M_8$ - 6.43%, $2MS_6$ - 5.6%, $2MN_6$ - 5.23%, $MO_3$ - 4.73%, $SN_4$ - 4.08%, $S_4$ - 0.92%, $2SM_6$ - 0.75%. These numbers were calculated based on a weighted sum of the harmonic amplitudes. Figure 8 shows the relative role of these higher harmonics compared to the M2 signal and the tidal map for M2 wave. In particular, Fig. 8a demonstrates the ratio of the M2 amplitude to the sum of amplitudes of the listed higher

harmonics. The role of nonlinear harmonics becomes significant in the interior zone of the domain, increasing with greater distance from the bottleneck and near the wetting/drying zone (Fig. 3a, Fig. 8). To quantify the role of the higher harmonics, Fig. 8b also demonstrates the tidal map for the M2 wave. The M2 wave amplitude is greatest in the southern part of the internal area. Also, we have one degenerate amphidromic point in the external part of the domain defined by capture of the Poincare wave.

The domain configuration and the foregoing results of the tidal energy transformation and evolution are signals of a pronounced tidal asymmetry in the tidal water level and in the current velocity behavior in the area. There are several reasons for the tidal asymmetry—in particular, the presence of the non-linear advection and bottom friction terms together with complex topography, resulting in a non-trivial wave interaction in the system (e.g., Friedrichs and Aubrey, 1988). The key geometric features directly impacting tidal distortion are the bathymetry relative to the tidal amplitude, the "bottleneck"

width and its variation during the tidal cycle, and the area occupied by the intertidal zone and its distance to the main tidal inlets. For data about tidal asymmetry to inform sediment dynamics analysis, we concentrate on the ebb and flood durations as well as the mean and maximum velocities relations. Ebb is defined as a period when the water level at the current location is decreasing, and flood as the period when it is increasing. Figure 9a (left panel) represents the ratio between the maximum velocities during spring ebb and flood. Figure 9b (left panel) represents the ratio of mean velocities during ebb and flood

periods. Figure 9c (left panel) represents the ratio of mean ebb and flood durations. The analysis reflects near-bottom velocities; however, the pattern of ratios is nearly the same for the depth-averaged solution. We note generally that the near-bottom solution provides a more pronounced ebb or flood dominance. The features represented in Fig. 9 (level panel) show the *mean pattern* for the full tidal cycle (29.5 days). In such a domain, we can expect that the flood-ebb dominance feature is not constant but can change within a full tidal cycle. For some ebb-flood cycles, the picture presented in Fig 9 (left panel)



can be significantly different. Therefore, we have decided to calculate how often flood or ebb dominance takes place within a
full tidal cycle in terms of velocities and duration.

Figure 9 (right panel) comprises 57 floods and ebbs. (We considered 28.5 days, but the "first" and "last" flood and ebb of the
full tidal cycle—29.5 days—were removed to make sure that we were considering the beginning of the flood and ebb.) Then
we calculated how often the mean and maximum velocities of the ebb are larger during the following flood (Fig. 9a, b right

panel). We also calculated how often the ebb duration was smaller than that of the following flood (Fig. 9c, right panel). In
other words, Fig. 9a (right panel) shows for example the frequency of the maximum velocity during ebb being larger than
that of the following flood; a value of 57 means that across all 57 cycles, maximum velocities during ebb are larger than
during flood. From all these pictures, we have removed subareas which did not take part at least at one flood-ebb cycle.

Figure 9 was generated from simulations on the first grid. But an important result is that the difference between the ratios

(asymmetry indicators) on either grid touches only a few details, but not a general pattern. The maximum disagreement
occurred during comparison of the ratios of velocities, going up to 0.2 for ratios of mean and maximum velocities. However,
we would stress that the position of the ebb/flood dominance areas are the same with only a small difference at the edge of
the zones.

The patterns presented in the left and right panels show some common features, and this is to be expected. For example, if

the ebb velocities are larger than the flood velocities nearly every ebb-flood cycle, then the **averaged** value of the velocity
ratios will be large, and the corresponding figure in the right panel will show a high frequency of the ebb-dominance event.
Figure 9 (right panel) shows that there are a lot of zones that may behave differently during different periods of a full tidal
cycle. (The frequency is between 0 and 57.) Note also that Fig. 9a and 9b represent largely different patterns than Fig. 9c.
For, example, if the mean and maximum velocities in the internal part of the domain are larger during flood, it does not mean

that the flood will be shorter. There can be no mass conservation for the ebb-flood cycle for the particular area subunit. Note
that the ebb in the whole domain under consideration is typically shorter than the flood, except in the south-western part of
the internal area (Fig. 9c, left panel).

Figure 9 demonstrates that it is not correct to regard ebb or flood dominance based only on velocity characteristics or typical
flood/ebb durations when these characteristics can yield opposite answers. Additionally, these characteristics are not

necessarily the same within one full tidal cycle (29.5 days).

To explain the complexity of the presented pattern, we have split the domain under consideration into four parts designated
as zones 1 through 4 (Fig. 9a, left panel).

### 4.5.1 Zone 1

Zone 1 represents an area where we observe a progressive Kelvin wave (Fig. 5, 9). The flood-dominated (orange-colored)

zone, in terms of maximum velocities (figure 9a, left panel), matches the area where we have a comparatively small residual





circulation (Fig. 6). Gravity waves traverse the shallow zone nondispersively at a speed defined by $\sqrt{gH}$ (g is acceleration due to gravity, H is the full depth); so we suggest that the crest of the tide overtakes the trough here (e.g., Dronkers, 1986; Hayes, 1980; Saloman and Allen, 1983). This is confirmed by Fig. 9a (right panel), which shows that maximum velocities in this zone are nearly always larger during flood. Figure 9b (left panel) shows that mean velocities in this subzone are nearly

equal to each other in terms of the mean, with slight ebb dominance. However, also some tidal cycles exhibit larger mean flood velocities than mean ebb velocities (Fig. 9b, right panel).

### 4.5.2   Zone 2

In zone 2, we have an extensive intertidal zone and maximum amplitudes of the semidiurnal tides compared to other zones; also, this zone lies far from the Lister Deep area. In this zone, the role of higher harmonics is greatest (Fig. 8), revealing the

major role of non-linear friction effects and non-linearity in changes of the water-layer thickness: here, the ratio of tidal amplitude to depth is greatest (increasing from Lister Deep toward the southern part of the zone). This is also can be seen in Fig. 7, which shows M4 locked in a velocity phase of -90º to 90º relative to M2. As a result, water level changes propagate more slowly during ebb (Dronkers, 1986), and the time-delay between low water at the inlet to Lister Deep versus in the considered area is greater than that for high water. In every regard, the father away from Lister Deep, the more pronounced

the flood dominance becomes. This means that ebb duration increases, and velocities are larger during flood.

### 4.5.3 Zone 3

In zone 3 we have very patchy dynamics in terms of velocities during flood and ebb; however, it is typical for the whole area that ebb is shorter than flood (flood duration is between 1.1 and 1.3 of ebb duration). Lister Deep can be characterized generally by higher mean and maximum velocities during ebb; but some side channels can be characterized by higher

velocities during flood. As the tide turns at low water, strong currents are still flowing seaward out of the main ebb channel. As the water level rises, the flood currents seek the path of least resistance around the margin of the delta. This creates horizontal segregation of flood and ebb currents in the tidal channels for a time. This is apparent in Fig. 10, depicting the transitional moment from ebb to flood.

In this zone, we emphasize subareas A and B as prime examples of the subdomains where mean currents ratio and maximum

currents ratio (Fig. 9a, b) are not synchronized with the durations ratio (Fig. 9c). For example, Fig. 9a and 9b demonstrate that the velocities at B are larger during flood in terms of the mean and maximum; but flood duration is longer than ebb. This signals that the water parcels are travelling different pathways during flood and ebb.

### 4.5.3   Zone 4

Zone 4 represents a small, semi-enclosed bight with a flood-dominated main channel and an ebb-dominated area around, and

in the inner part of the bight in terms of velocity. However, we would stress that flood dominance, though typical, is not a constant feature. For several cycles within the full tidal cycle (29.5), the mean and maximum velocities will be larger during





ebb. For the whole zone, ebb is shorter than flood for almost the full tidal cycle (29.5 days). The reason the pattern is opposite to that of zone 3 lies in the small volume of intertidal storage and large variation in the "main channel" width. In this zone, frictional drag is insignificantly greater at low water than at high water. Also, the role of M4 tide compared to M2

tide is greatest there (not shown); the role of the sum of all non-linear harmonics is smaller than in zone 2 (Fig. 8). For this zone, ebb and flood generally start on the sides of the main channel Rømø Deep in marginal ebb-dominated channels.

## 5 Discussion

### 5.1 Bedform peculiarities

The most interesting question for future study is how the given asymmetry and residual circulation pictures line up with the

bedform peculiarities. Detailed analysis of these processes requires a coupled sediment module as well as wind and wave forcing. However, as we have shown (Table 2, Fig. 2) that tides can explain a large part (more than 80 %) of the current velocities in the area of Lister Deep, some prognoses can be made.

The results presented here suggest the presence of subaqueous dunes with stable characteristics in areas where mean and maximum current velocities are permanently higher or lower during ebb/flood than during flood/ebb. With this definition

(see Fig. 9), our results agree with those presented in Boldreel et al. (2010) and Mielck et al. (2012). In these studies conducted in the working area Lister Deep and adjacent to Königshafen, dunes of various sizes, escarpments and other erosional features were analyzed based on hydroacoustic data (seismics, sidescan sonar and RoxAnn seafloor classification system) to determine dune characteristics and orientations (and hence flood or ebb dominance) of the particular areas. Boldreel et al. (2010) state that the flood-dominated dunes are larger than ebb-dominated dunes in the area of the bottleneck.

Our study reveals that, in this zone, the ebb is generally shorter than the flood (Fig. 9c), which may explain this observation. The map of the residual circulation (Fig. 6) suggests probable directions of subaqueous dune migration where bottom currents are strong enough.

### 5.2 Grid performance

Each presented grid had advantages and disadvantages. The first grid, the curvilinear one, offered minimum numerical

viscosity but was not very flexible when it came to choosing grid cell size as compared to an unstructured grid. This is especially crucial in the zones of large bathymetric gradients and in the wetting/drying zones. The second grid is more dissipative and more sensitive to the quality of the open boundary solutions. We can conclude that the first grid would be most suitable for studying baroclinic dynamics in the area, while the second grid will be most suitable for sediment dynamics.



## 6 Summary

This study is dedicated to tidally-induced dynamics in the SRB, with a focus on the non-linear component. The newly obtained high-quality bathymetric data supported the use of high-resolution grids (up to 2 meters in the wetting/drying zone) and elaboration of the details of tidal energy transformation in the domain. The FESOM-C model was used as the numerical tool. In preparation, different open boundary conditions for the summary tide (major diurnal and semidiurnal as well as M4

constituents) were tested for the best fit with observational data. The simulations with open boundary conditions from TPXO 9 showed very satisfying agreement with available observations. Based on 3D barotropic runs, the ellipses, energy fluxes, residual circulation and tidal asymmetry maps were constructed and analyzed for the whole area for the first time. Four zones with fundamentally different asymmetry structures were indicated during tidal asymmetry analysis. We also showed that it is not correct to talk about ebb or flood dominance based only on one velocity characteristic or typical flood/ebb

durations since these indicators can yield an opposite answer. Also asymmetry indicators were not constant everywhere from one ebb-flood cycle to another.

All experiments were performed on two grids with different structures and resolution details. Energy balance fulfillment was shown for both grids. The generated maps generally showed the same pattern on both grids, which allowed us to conclude that further increasing the resolution will not lead to pronounced changes in the results. The obtained results are a necessary

and useful benchmark for further studies in the area, including for work on baroclinic and sediment dynamics. It would be fruitful to pursue further research about how the obtained maps reflect bedform peculiarities in the area in order to predict bedform characteristics in hard-to-reach places.

**Author contribution**

VF designed the experiments and carried them out, prepared the second grid and new bathymetry product, suggested

asymmetry analysis. AA constructed first grid, encouraged FV to investigate some aspects of the nonlinear dynamics, wrote the part about tidal ellipses and helped to visualize the results. LS carried and provided the multi-beam data, participated in the construction of the new bathymetry product and consulted VF a lot during the verification stage. IK and HCH discussed with VF all experiments and results providing valuable comments and remarks. FA carried and processed ADCP data, prepared the reference list and together with HCH and LS significantly improved the manuscript readability. HCH provided

the information about bedform peculiarities in the area. KHW supervised the research. All authors contributed to the final manuscript, provided critical feedback and helped to shape the research.

**Competing interests**

The authors declare that they have no conflict of interest.



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





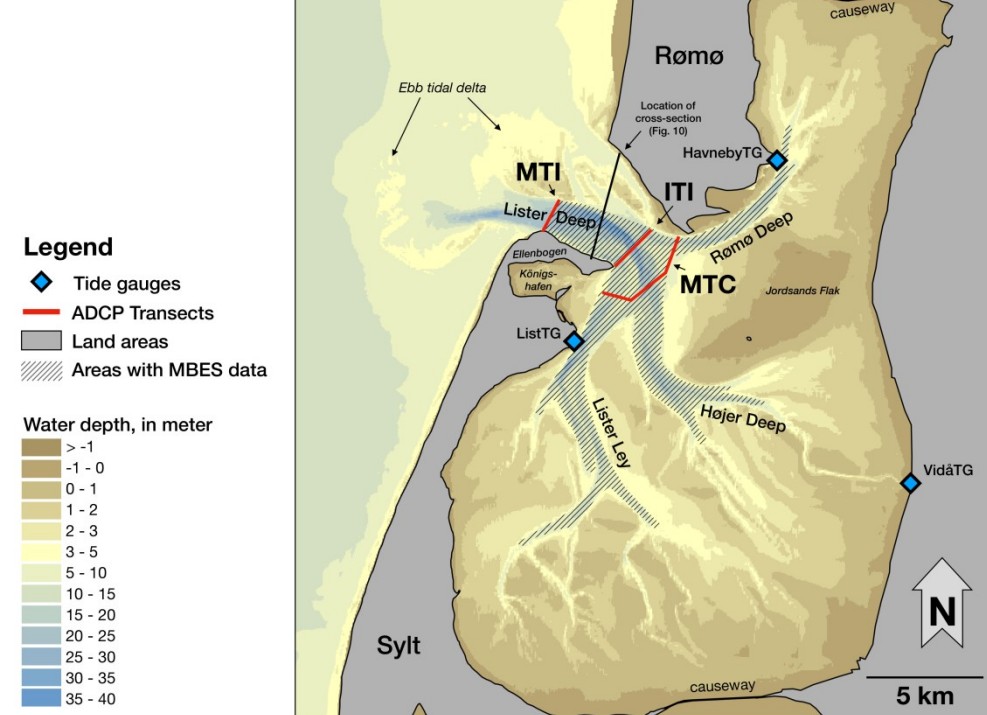

**Figure 1.** The domain under consideration and its bathymetry. The red lines indicate ADCP transect positions, the blue rhombuses show positions of the tide gauges. The shaded area represents zone with the resent multibeam echo sounder (MBES) data.






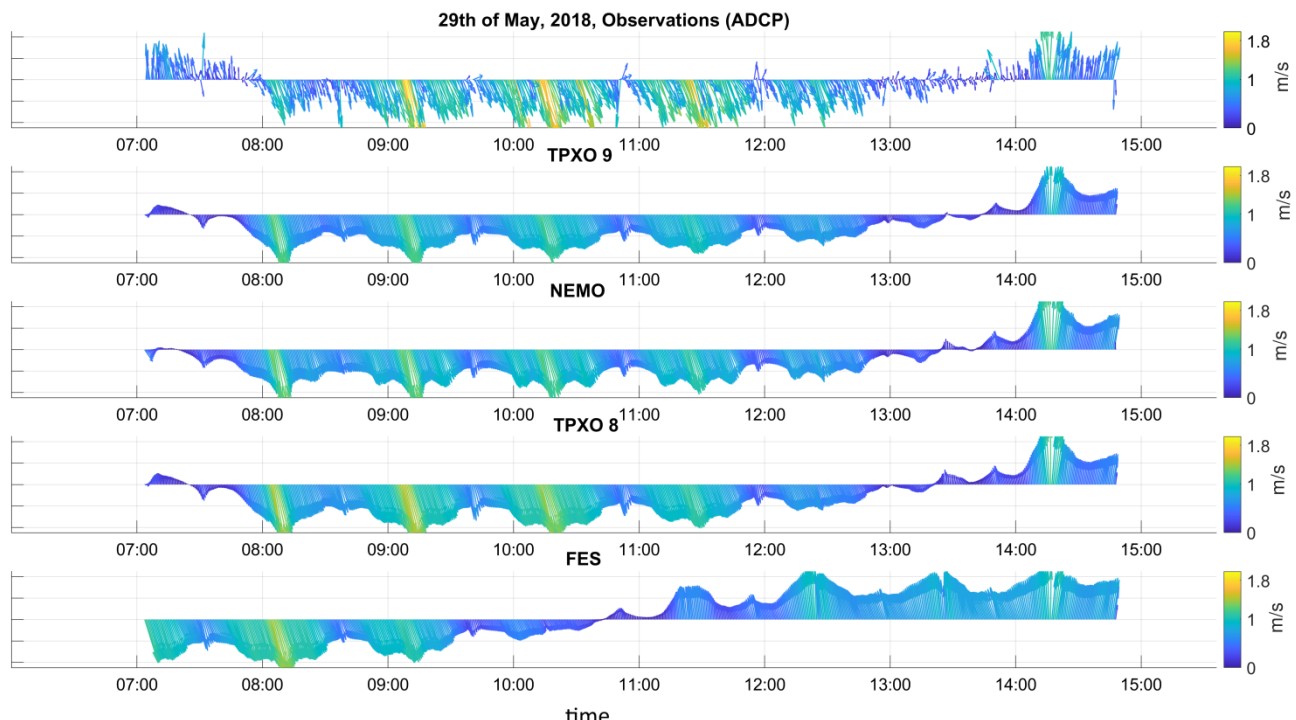

**Figure 2.** The observed and modelled depth-averaged velocities on May 29. The arrow color indicates the velocity magnitude [m/s]; arrow directions indicate the velocity direction.





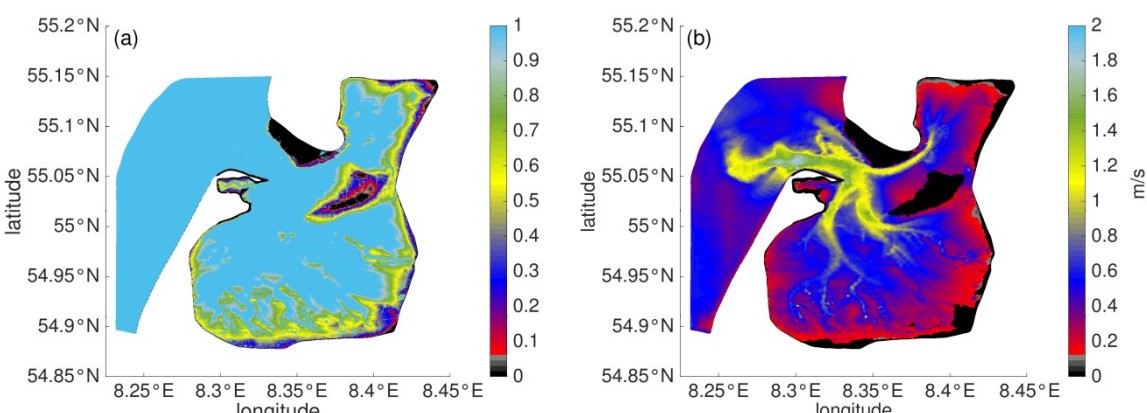

**Figure 3.** a) The probability of wetting based on simulation results for the two full tidal cycles using the second grid. b) The maximum tidal velocities at the current position using the second grid.

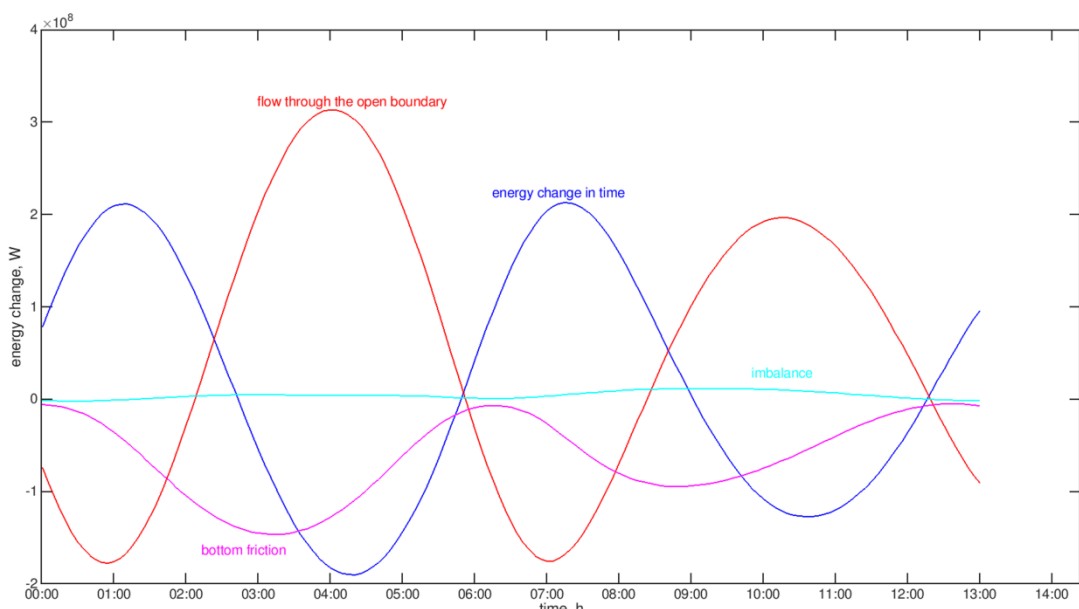

**Figure 4.** The energy budget for the depth-averaged solution with open boundary conditions from the TPXO 9 database for summary tide, [W], in blue: energy change in time, in red: flow through the open boundaries, in magenta: bottom friction, in cyan: imbalance.




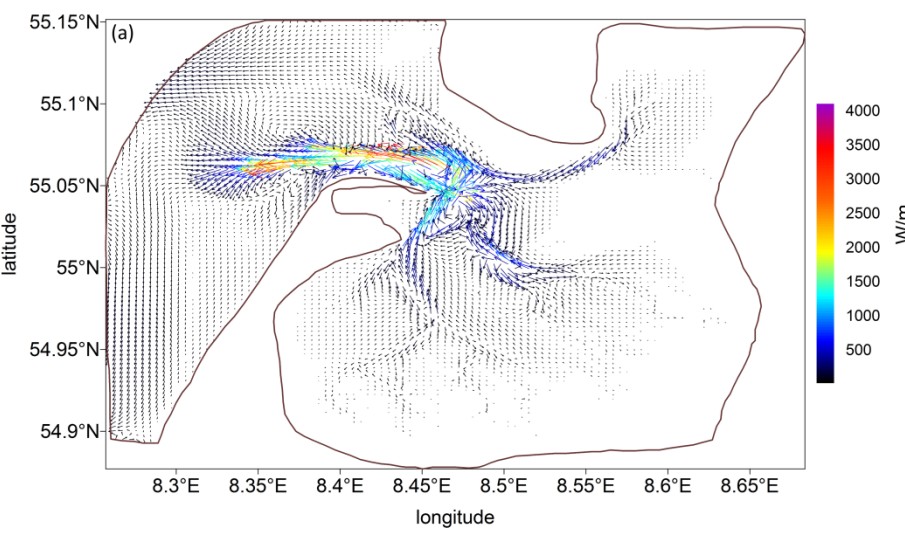


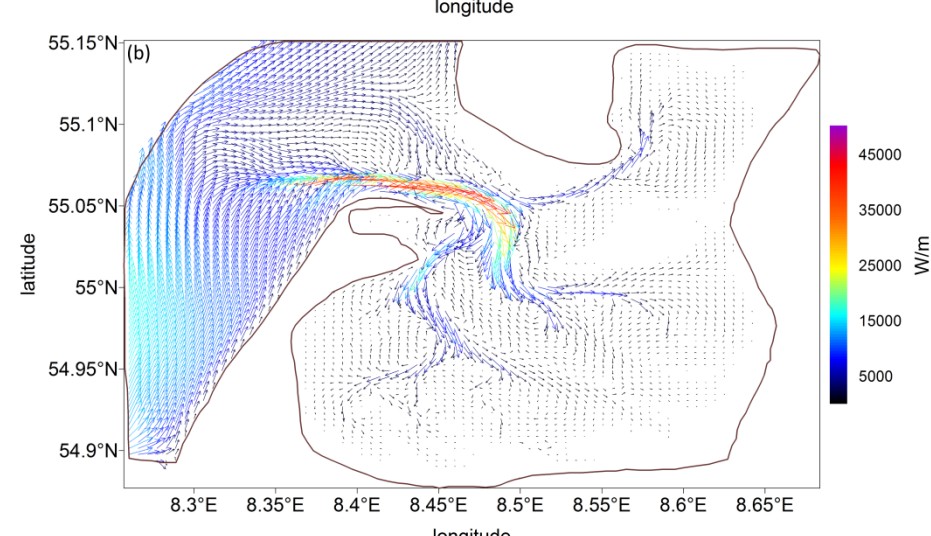





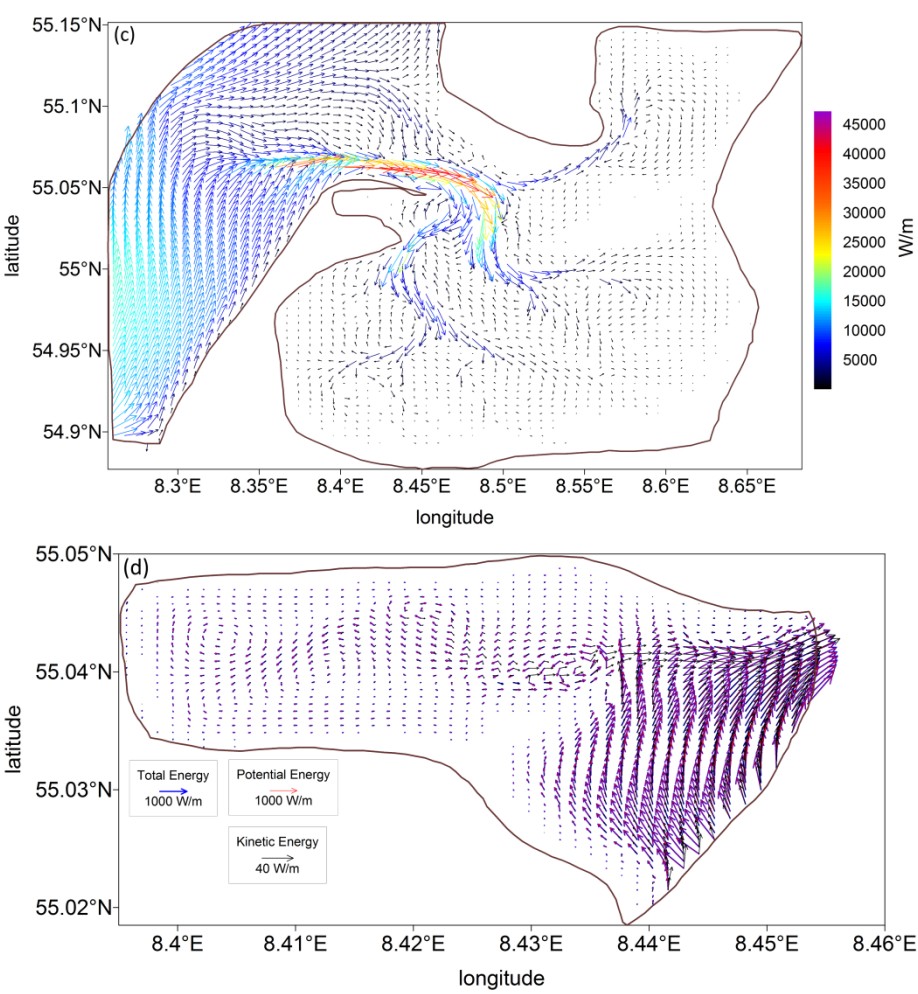

**Figure 5.** The flux of tidal energy (summary tide): a) kinetic; b) potential; c) total ($\mathbf{E_\lambda}$, $\mathbf{E_\theta}$); d) for the Königshafen subarea. The length of vectors on the maps is scaled based on the square root method.





**Figure 6**. Residual circulation of the summary tide for: a) the whole area considered; b) the Königshafen embayment. The '+' and '-' illustrate the clockwise and counterclockwise rotation of the major gyres present in the system. The residual circulation is demonstrated on the second grid.



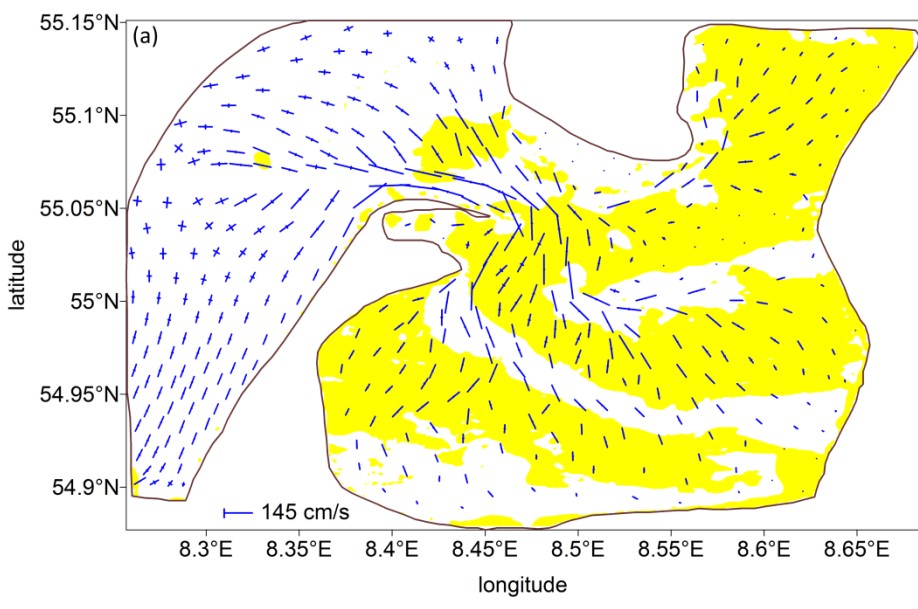


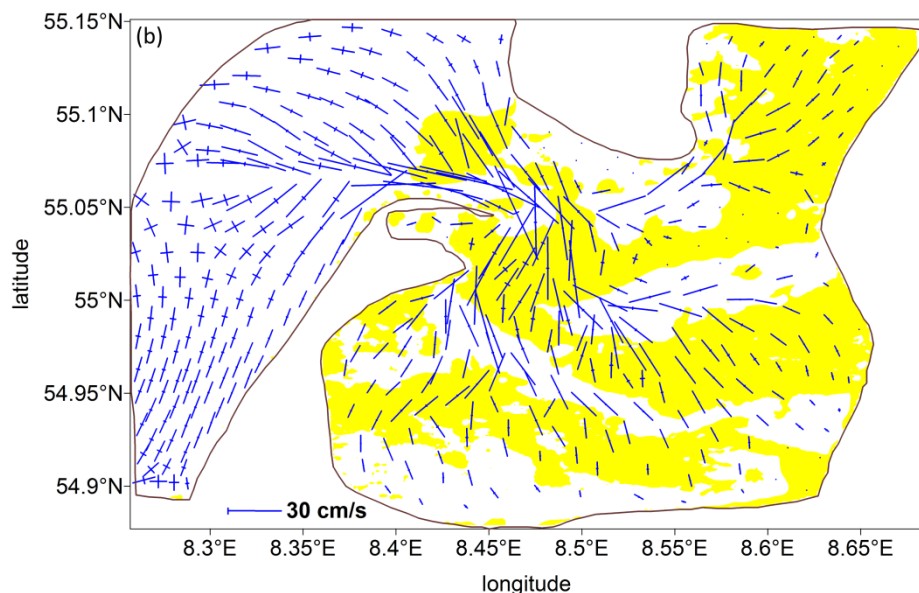



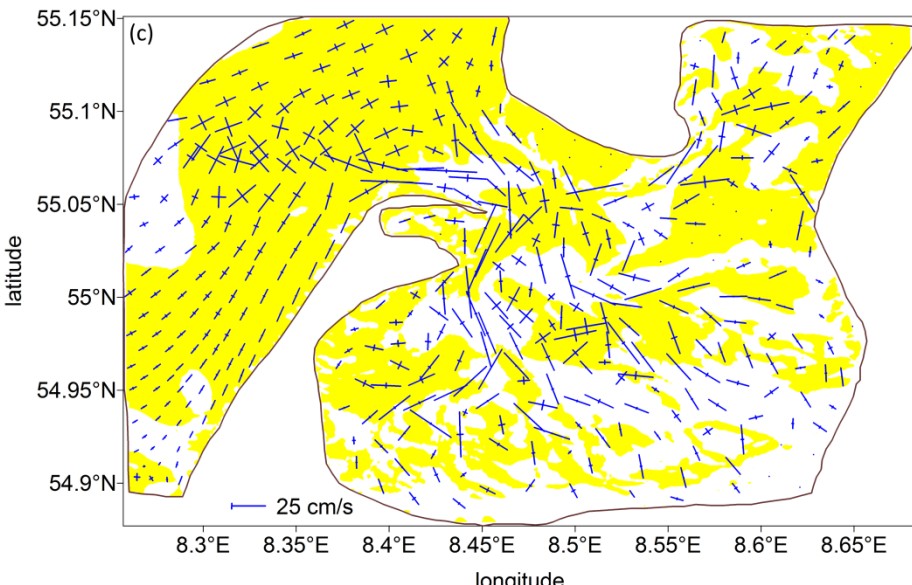

**Figure 7.** Axes of the a) M2, b) S2, c) M4 current ellipses and zones of the minor-to-major axis ratios. Yellow zones denote the domain of anticyclonic rotation of the current vector.








**Figure 8**. a) The relative weight of the nonlinearity: the ratio of M2 wave amplitude to the sum of the major nonlinear
constituents. b) Tidal map for the M2 wave: phases are demonstrated via contour lines, º, amplitudes [m] are via color bar.
The dark blue color indicates zone, where topography features are above sea level.

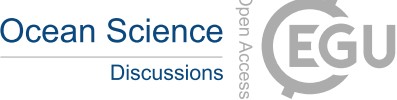






**Figure 9.** The ebb-flood dominance asymmetry maps. Left panel: ratio of a) maximum velocities during ebb and flood; b) mean velocities during ebb and flood; c) durations of flood and ebb. Right panel: frequency of event within full tidal cycle a) maximum velocities during ebb are higher than during flood; b) mean velocities during ebb are higher than during flood; c) flood is longer than ebb. Numbers 1-4 indicate different zones under consideration. The A, B demonstrates an example of the subareas in which the strong flood dominance in terms of velocity does not mean shorter flood. Subareas which do not take part at least at one flood-ebb cycle are removed.







**Figure 10.** Snap-shots of direction relative to the y-axes (counted positively counterclockwise), [º], and magnitude [m/s] of velocities within the cross-section which connects Sylt (left shore) and Rømø (right shore); the angle between y-axes and the cross-section is -16º . In the left panel, the grey-green color indicates water outflow; blue-red color indicates water inflow. The snapshots capture the transition phase from ebb to flood with a 20-minute interval from one picture to another (top to bottom). The top panel is a snapshot of a moment when the external part of the domain is already at first-stage flood and the internal part of the domain is still at last-stage ebb.





**Table 1** - The summary of the five cruises on board of Research Vessel "Mya II" profiling three main transects: the Inner Tidal Inlet (ITI), the Main Tidal Channels (MTC) and the Outer Tidal Inlet (OTI).

| Transect Name | Date | Lon / Lat (°) (Start – End) | Tidal Period | Duration (h) |
|---|---|---|---|---|
| ITI | 22/05/2018 | 8.464 / 55.047 <br> 8.489 / 55.062 | neap | 6:28 |
| MTC | 23/05/2018 | 8.461 / 55.039 <br> 8.474 / 55.035 | neap+1 day | 6:19 |
|  | 24/05/2018 |  | neap+2 days | 5:56 |
|  |  | 8.499 / 55.046 <br> 8.505 / 55.056 |  |  |
| ITI | 29/05/2018 | 8.464 / 55.047 <br> 8.489 / 55.062 | spring | 7:11 |
| OTI | 30/05/2018 | 8.418 / 55.059 <br> 8.428 / 55.071 | spring+1 day | 12:24 |





720

**Table 2**. The inter-comparison of the observed and simulated velocities based on different open boundary conditions in the area of Lister Deep. The table presents Root Mean Square Deviation (RMSD, m/s) and correlation coefficients (C.C.) for 'u' and 'v' velocity components. The results are given for the simulations performed on the first grid.

| N of obs. | Date in May | | FES2014 | NEMO | TPXO8.1 | TPX09 | Mean vel., m/s (obs.) | Max vel., m/s (obs.) |
|---|---|---|---|---|---|---|---|---|
| 655 | 22 | RMSD | 0.34 | 0.34 | 0.3 | 0.28 | 0.64 | 1.43 |
| | | C.C. u, v | 0.84, 0.83 | 0.83, 0.83 | 0.84, 0.84 | 0.87, 0.85 | | |
| 637 | 23 | RMSD | 0.32 | 0.35 | 0.31 | 0.31 | 0.55 | 1.56 |
| | | C.C. u, v | 0.81, 0.82 | 0.78, 0.78 | 0.81, 0.82 | 0.82, 0.82 | | |
| 618 | 24 | RMSD | 0.41 | 0.32 | 0.34 | 0.31 | 0.54 | 1.31 |
| | | C.C. u, v | 0.69, 0.5 | 0.83, 0.64 | 0.8, 0.62 | 0.84, 0.66 | | |
| 764 | 29 | RMSD | 0.92 | 0.32 | 0.31 | 0.3 | 0.67 | 1.98 |
| | | C.C. u, v | 0.09, 0.26 | 0.86, 0.84 | 0.86, 0.85 | 0.89, 0.87 | | |
| 1259 | 30 | RMSD | 1 | 0.33 | 0.26 | 0.26 | 0.8 | 1.73 |
| | | C.C. u, v | 0.18, 0.13 | 0.94, 0.76 | 0.97, 0.77 | 0.97, 0.76 | | |





725

**Table 3.** Simulated and observed amplitudes and phases of the major tidal constituents at different locations.

| | First grid | | Second grid | | Observations | |
|---|---|---|---|---|---|---|
| | **List TG** | | | | | |
| | Amp | Phase | Amp | Phase | Amp | Phase |
| **M2** | 75.44 | 23.04 | 74.2 | 23.48 | 77.92 | 23.90 |
| **S2** | 17.12 | 85.01 | 19.34 | 95.83 | 18.92 | 94.55 |
| **N2** | 12.38 | 8.51 | 11.56 | 10.67 | 13.26 | 358.24 |
| **O1** | 6.74 | 277.2 | 6.77 | 276.81 | 8.33 | 269.30 |
| **K1** | 5.21 | 54.1 | 3.78 | 46.6 | 5.80 | 56.70 |
| **Q1** | 1.84 | 210.28 | 1.8 | 211.86 | 2.64 | 210.81 |
| **M4** | 4.92 | 188.68 | 3.89 | 181.43 | 5.08 | 214.86 |
| **RMSD: amp(cm)/ ph(°)** | 1.4 / 11.7 | | 1.9 / 14.3 | | | |
| | **Havneby TG** | | | | | |
| | First grid | | Second grid | | Observations | |
| | Amp | Phase | Amp | Phase | Amp | Phase |
| **M2** | 76.23 | 24.21 | 75.26 | 24.91 | 79.83 | 24.78 |
| **S2** | 17.31 | 85.92 | 19.95 | 97.28 | 19.53 | 94.88 |
| **N2** | 12.49 | 9.23 | 11.87 | 11.95 | 13.42 | 359.15 |
| **O1** | 6.82 | 278.98 | 6.80 | 279.01 | 8.25 | 270.42 |
| **K1** | 5.26 | 56.12 | 3.80 | 50.54 | 5.90 | 62.26 |
| **Q1** | 1.83 | 211.65 | 1.80 | 213.16 | 2.85 | 211.03 |
| **M4** | 7.28 | 192.79 | 7.04 | 196.26 | 7.84 | 224.89 |





| RMSD: amp(cm)/ ph(°) | 1.8 / 13.75 | | 2.1 / 13.12 | | | |
|---|---|---|---|---|---|---|
| **Vidå TG** | | | | | | |
| | **First grid** | | **Second grid** | | **Observations** | |
| **M2** | 67.74 | 35.94 | 65.71 | 37.21 | 63.41 | 42.35 |
| **S2** | 13.26 | 105.05 | 16.06 | 114.17 | 14.89 | 113.10 |
| **N2** | 9.82 | 29.47 | 8.91 | 31.19 | 9.78 | 18.30 |
| **O1** | 6.13 | 294.36 | 6 | 295.47 | 6.95 | 284.53 |
| **K1** | 4.9 | 73.47 | 3.59 | 71.78 | 5.58 | 79.42 |
| **Q1** | 1.19 | 223.36 | 1.25 | 224.92 | 1.78 | 217.68 |
| **M4** | 2.39 | 90.38 | 3.21 | 55.18 | 4.77 | 25.87 |
| RMSD: amp(cm)/ ph(°) | 2 / 25.5 | | 1.5 / 13.5 | | | |

730