# Peer review of "Non-linear aspects of the tidal dynamics in the Sylt-Rømø Bight, south-eastern North Sea"

_Ocean Science, 2019_

## Referee Comment (RC1) · Anonymous Referee #1 · 23 Aug 2019

This paper focuses on tidal dynamics in the Sylt-Rømø bight as modeled by the unstructured grid FESOM-C model. The model has a high spatial resolution and bathymetry. The paper discusses effects of model spatial resolution on the circulation results and outlines characteristics of tidal dynamics in the basin.

The application and results are interesting and as such I would recommend the paper to be published. However, the manuscript needs to be improved; most notable the model setup and used methodology should be described more accurately to allow reproducibility.

Specific comments

Section 2.2: It would be useful to include a figure of the meshes to a) see the exact

location of open boundary and b) details of the mesh resolution in crucial areas (e.g. Lister Deep, shallow areas). As the grid plays an important role in such studies, the authors should elaborate on the criteria used to choose the mesh resolution.

line 103: Which turbulence closure model are you using? E.g. k-epsilon, or one of the Generic Length Scale models that GOTM provides?

Section 2.3: The reference to the NEMO model is inaccurate. I presume that in this study the European north-west shelf model is used and it should be cited appropriately. Why do you compute tidal harmonics from the shelf model results? This potentially introduces an error source; it would be better to use the elevation time series itself as it includes atmospheric effects.

Open data: Are the bathymetric and ADCP data sets introduced in this manuscript publicly available?

Section 2: The model configuration should be elaborated. What were the calibration and analysis period(s)? What were the initial conditions? Was there a spin-up period? It appears that open boundary conditions were calibrated with a 2D model using bottom drag parametrization (lines 224-226), while subsequent results were carried out with a 3D model. For clarity, please define these configurations.

Harmonic analysis: The observations of velocity and water elevation observations do include atmospheric effects which are absent in the tidal models. What were the atmospheric conditions during the calibration and validation periods? Can you estimate their impact on the error metrics? Are the tides in this small-scale system really stationary so that harmonic analysis is well defined?

line 195: Is Cd constant in space? If so, is that a realistic configuration for the bight?

Tide gauge comparisons: It would be useful to have example time series comparing the observations and the model to give a better idea of the model's performance. A Taylor diagram could also be used.

line 219: Do you use a sponge layer? If so, please describe it in Section 2.

Baroclinicity: The tidal dynamics, e.g. tidal ellipses, are probably affected by density and stratification effects which in the present study is neglected. Can you argue that baroclinic effects are negligible in this system?

Section 5.2: Larger dissipation in the unstructured grid could also be due to better resolved intertidal dynamics that are inherently dissipative. Presumably also the bathymetric features are quite different in these two grids.

The authors conclude that the model results "converge" to a realistic solution (line 81, 478, abstract), based on the presented simulations with two different grids. The authors also conclude that the curvilinear grid has lower dissipation making it thus better suited for baroclinic studies. I find these conclusions somewhat premature: Only two different grids were used, which at the same time had different element types (triangles and quads), resolution and topology (unstructured and curvilinear), as well as (I presume) bathymetry. As such, it is really quite hard to infer what grid properties cause the observed change in model performance. The grid sensitivity study should be extended to better address the effects.

Technical corrections

line 123: 1/30 *degree*

line 182: this paragraph is duplicate of the previous one.

Figure 3 a: The ratio being thus defined it would be more appropriate to call it "weight of linearity" instead. The unit (m) is wrong.

Table 3: Add units. What do "RMSD", "amp(cm)", "ph(°)" stand for?

Figure 9: Add units.

---

## Referee Comment (RC2) · Anonymous Referee #2 · 25 Aug 2019

General comments:

The article is a detailed study of the nonlinear dynamics of the tide in the shallow bay of the North Sea, which is characterized by a significant area of the intertidal zone. As a research tool, a relatively new FESOM-c model is used, which approximates the governing equations by the finite volume method and is able to work on hybrid unstructured computational grids. New data on the bathymetry of the bay and tidal currents are also presented. The authors of the study set a rather difficult goal, analyzing the results of observations and modeling for the summary tide instead of effects of nonlinearity for the single harmonic tide. In the latter case, it would be easier to estimate the contribution of the main mechanisms of nonlinearity (shallow depth, advection, and quadratic friction) on the structure of tidal asymmetries. Nevertheless, the results pre-

sented in the article are of real interest, since they provide a serious basis for analyzing the features of sediment dynamics and the formation of a variable bottom relief. I would recommend this paper to be published after a little revision. I hope the comments below help improve the manuscript.

Specific comments:

Line 100: Are only 10 sigma vertical layers enough for 3D simulations? In other words, is the numerical solution dependent on the number of vertical layers?

Section 2.3 (Open boundary conditions) Please specify the period of model calculations (dates).

Line 165 (Data. 3.2 Tide gauge (TG) data): VidaTG station is located in the intertidal zone, and so during low tide ( when the seabed is exposed) the continuous (quasi harmonic) time series is greatly distorted. (see for example, https://www.emodnet-physics.eu/map/platinfo/piroosplot.aspx?platformid=9015&60days=true). In fact, this is time series with data gaps. Nevertheless, the authors used classical harmonic analysis for this station in the validation of the model. Did authors take into account the peculiarity of the tide in this station when analyzing the results? (see also the comment below on Section 4.5.2).

Table 2. The results of the inter comparison show rather large RMSD values (compared with the velocity values themselves). Apparently, this is due to the neglect of wind-induced fluctuations in the simulation. The question arises: Is inter comparison in Table_2 appropriate?

Line 330: It seems to me that the use of the term "seiche" is not entirely appropriate in this context, since we are dealing with forced fluctuations. It would be more correct to speak about oscillations as near standing wave.

Line 350: It is desirable to immediately emphasize that the results of Fig. 8 relate to the sea level (not currents).

Line 359: Frankly, I do not see indication on degenerate amphidromic point in external area. Yes, there is a slight closeness of phase contours lines (caused by proximity to the strait and refraction of the tidal wave), but there is no decrease in the tide amplitude as it is usually in nodal (amphidromic) zone. The effect of the capture of a Poincare wave is interesting, but requires explanation or reference.

Line 365-370: It is interesting how the authors distinguish between the different duration of the ebb and tide caused by the nonlinearity from the effect of the sum of harmonics of various periods. The last effect is called the Diurnal Inequality of tide, and it is not connected with non-linearity.

Line 405-410: To explain the effect of the dominance of tidt velocities over ebb ones, it is not necessary to use phase speed inequality (SQRT (gh)). There is a simple explanation for this effect: the bottom friction dampens the flow more efficiently in smaller depth (during low ebb).

Section 4.5.2 (Line 412). Again about the accuracy of harmonic analysis in the intertidal zone. Indeed, in this zone, at low tide, the bottom is exposed, and this means that data gaps appear in the model time series. In this case, the classical harmonic analysis can give inaccurate results for the amplitudes and phases of the waves. It seems that it would be more logical to exclude intertidal zones from the analysis of results (at least in terms of the results of harmonic analysis). How was this fact taken into account? Otherwise it's necessary apply special procedures to avoid mistakes.

Section 6 (Summary). Unfortunately, the question of the sensitivity of the numerical simulation to the accuracy of bathymetric data remained outside the discussion. In particular, how important is the effect of new bathymetric data in the strait on simulation results. A comparison of the solutions with the old and new bathymetry would answer this question, and perhaps provide a serious justification for the need for new good bathymetry for the whole area.

Technical corrections:

Lines 175 and 182: Two paragraphs contain the same information.

Line 245: 29.5 days is rather a lunar (synodic) month than a tidal cycle

Line 335: Direction of rotation (not orientation) of the tidal ellipse is determined by the sign of the ellipticity. (Orientation is rather the inclination of its main axis of the ellipse).

Line 372: Replace "(level panel)" by "(left panel)"

Figure 5. Please explain what the following sentence means: "The length of vectors on the maps is scaled based on the square root method."

Figure 8 .: "The dark blue color indicates zone, where topography features are above sea level." ....... above the highest sea level ???

---

## Editor Comment (EC1) · Mattias Green (Editor) · 13 Sep 2019

The authors use the high resolution FESOM-C to investigate the non-linear tides in a part of the Wadden Sea. Using a new batymetry, the tidal dynamics and energetics was described for the area. The paper is an insteresting local study, and the methods and analysis will eb of interest for other investigations. The implications of the work is also interesting, e.g., in terms of seiment tranpsort and erosion. I agree with the reviewers' comments that the paper needs some more work, and i recommend a minor revision.

The reviewers provide a series of relevant comment that i agre with, and i have nothing major to add on my own. In particular, i think the model description needs more detail,

as suggested by both reviewers.

Also, the Vida TG station is dry at spring low - how is this taken into account? A traditional harmonic analysis on gappy data is not particularly reliable. Please comment.

The other comments from teh reviewers should be taken into account in the revised version.
* * *

---

## Author Comment (AC1) · 8 Oct 2019

Dear Referee 1,

I am very grateful to you for the valuable and useful comments and remarks! Please, find in the supplement the answers to the comments and remarks and updated version of the manuscript (the changes are marked in yellow).

Kind regards, Vera on behalf of the co-authors

Please also note the supplement to this comment:
https://www.ocean-sci-discuss.net/os-2019-73/os-2019-73-AC1-supplement.zip

---

## Author Comment (AC2) · 8 Oct 2019

Dear Referee 2,

Thank you a lot for the valuable comments! Please, find in the supplement the answers to the comments and updated version of the manuscript (the changes are marked in yellow).

Kind regards, Vera on behalf of the co-authors

Please also note the supplement to this comment:
https://www.ocean-sci-discuss.net/os-2019-73/os-2019-73-AC2-supplement.zip

---

## Author Comment (AC3) · 8 Oct 2019

Dear Editor,

Thank you for the remarks and concern! Please, find our response together with up-dated version of the manuscript (the changes are marked in yellow) attached.

Kind regards, Vera on behalf of the co-authors

Please also note the supplement to this comment:
https://www.ocean-sci-discuss.net/os-2019-73/os-2019-73-AC3-supplement.zip
* * *

---

## Referee Report (RR1)

[referee-annotated manuscript omitted]

---

## Author Response (AR3)

**Referee 1**

Dear Referee 1,

I am very grateful to you for the valuable and useful comments and remarks! Please, find below the answers to the comments and remarks.

5  Kind regards, Vera on behalf of the co-authors

Section 2.2: It would be useful to include a figure of the meshes to a) see the exact location of open boundary and b) details of the mesh resolution in crucial areas (e.g. Lister Deep, shallow areas). As the grid plays an important role in such studies, the authors should elaborate on the criteria used to choose the mesh resolution

10  Thank you for the comment! Please, find an answer below.

The minimum grid cell size (2-14 meters) was chosen based on available bathymetry data resolution. In case of the curvilinear grid the cell sizes are determinate by the square of the subarea. The whole domain was divided into 6 subareas (Fig. 1a). The smallest among them are those, which are covered the Königshaven area and Lister zone. The division of the domain into subareas was dictated by the

15  bathymetry information and  area of particular interests - Königshaven, Lister Deep and ets. Figure 1 shows the subdivisions, grid quality and final nodal areas for the curvilinear grid.

[Figure]

Fig. 1. Demonstration of the curvilinear grid: a) subdivision into subareas; b) grid quality; c) nodal area, pink color indicates open boundary position.

[Figure]

Fig. 2. Demonstration of the unstructured grid: a) grid quality; b) grid scale matrix; c) nodal area, pink color indicates open boundary position.

In case of the mixed mesh (unstructured grid) we were more flexible in choosing of the grid sizes. The matrix of element sizes was constructed based on the information about the bathymetry, the bathymetry gradient and the zones of particular interest (Lister Deep; main draining channels). Figure 2 shows the grid quality and nodal area.

Both grids have nearly the same open boundary position.

The additional figures have been added to Appendix, the additional notes have been added to the text of the manuscript.

line 103: Which turbulence closure model are you using? E.g. k-epsilon, or one of the Generic Length Scale models that GOTM provides?

The k-epsilon closure was used. The information has been added to the text.

Section 2.3: The reference to the NEMO model is inaccurate. I presume that in this study the European north-west shelf model is used and it should be cited appropriately. Why do you compute tidal harmonics from the shelf model results? This potentially introduces an error source; it would be better to use the elevation time series itself as it includes atmospheric effects. Open data: Are the bathymetric and ADCP data sets introduced in this manuscript publicly available?

Yes, the European north-west shelf model results were used, the ocean part of which is based on NEMO. We have completed the reference by citation of the web-source with full description of the setup: http://marine.copernicus.eu/services-portfolio/access-to-products/?option=com_csw&view=details&product_id=NORTHWESTSHELF_ANALYSIS_FORECAST_PHY_004_013

We did not take the elevation directly, because in this study we did not include the wind forcing. We took for the analysis the elevation time series for the whole year, therefore we hope that the analysis mistake is sufficiently small.

The ADCP data used in this study are available in PANGAEA: https://doi.pangaea.de/10.1594/PANGAEA.894070

The bathymetry data will be made publicly available this year or in the beginning of the next year and also will be available in PANGAEA database.

The additional information has been added to the text.

Section 2: The model configuration should be elaborated. What were the calibration and analysis period(s)? What were the initial conditions? Was there a spin-up period? It appears that open boundary conditions were calibrated with a 2D model using bottom drag parametrization (lines 224-226), while subsequent results were carried out with a 3D model. For clarity, please define these configurations.

All results demonstrated in the manuscript are obtained based on barotropic simulation with tidal forcing only. To find optimal tidal solution at the open boundary we performed series of 2D experiments, mainly, because the intercomparion with observations was performed based on depth-averaged velocities. After that we moved to multi-layer task. The optimal roughness height was set to 0.001 m for the TPXO9 solution; this value agreed with one estimated from observations in a similar region (Werner et al., 2003) and, in terms of the mean, with the Cd equaled to 0.0025 for the 2D-scenario (we made series of sensitivity experiments to find this value). Table 1 presents the comparison of the RMSD and Correlation coefficient with the available ADCP data for 2D and 3D cases on 1st grid with Cd=0.0025 and r=0.001 m. Table 1 shows that 2D and 3D cases show similar results in terms of intercomparison with observations.

| N of obs. | Date in May | | TPX09, curvilinear grid, 3D case, $z_0=0.001$ | TPX09, curvilinear grid, 2D case, $C_d=0.0025$ |
|---|---|---|---|---|
| 655 | 22 | RMSD | 0.28 | 0.28 |
| | | C.C. u, v | 0.87, 0.86 | 0.87, 0.85 |
| 637 | 23 | RMSD | 0.31 | 0.31 |
| | | C.C. u, v | 0.82, 0.82 | 0.82 0.82 |
| 618 | 24 | RMSD | 0.31 | 0.31 |
| | | C.C. u, v | 0.84, 0.7 | 0.84, 0.66 |

| 764 | 29 | RMSD | 0.29 | 0.3 |
|------|-----|------|------|------|
| | | C.C. u, v | 0.89, 0.89 | 0.89, 0.87 |
| 1259 | 30 | RMSD | 0.25 | 0.26 |
| | | C.C. u, v | 0.97, 0.78 | 0.97, 0.76 |

70 The spin-up period for all simulations was three months with a criteria of the stabilization of the total energy behavior. Due to the fact that paper considers only the tidal dynamics for the analysis we took last two full tidal cycles - 59 days. We simulated the tidal dynamics in 2018, which is expressed in Doodsen correction of the prescribed amplitudes and phases, therefore we were able to compare velocities second to second.

75 The additional setup details have been added to the manuscript.

Harmonic analysis: The observations of velocity and water elevation observations do include atmospheric effects which are absent in the tidal models. What were the atmospheric conditions during the calibration and validation periods? Can you estimate their impact on the error metrics? Are the tides in this small-scale system really stationary
80 so that harmonic analysis is well defined?

Table 1. Summary of the five cruises on board of Research Vessel Mya II, profiling three main transects: Inner Tidal Inlet (ITI), Main Tidal Channels (MTC) and Outer Tidal Inlet (OTI). Main Tidal Channels (MTC) show the initial, turning and ending points, since it covers three sections in one transect. The Wind Roses characterize the wind conditions during cruise times, with the legend colors
85 representing the wind velocity in m/s and the circles representing the frequency percentage of the direction from where the wind blows.

| Transect Name | Date | Lon / Lat (°) (Start – End) | Tidal Period | Duration (h) | Wind conditions (Wind Rose) |
|---------------|------|------------------------------|--------------|--------------|------------------------------|

| | | | | | | |
|---|---|---|---|---|---|---|
| ITI | 22/05/2018 | 8.464
55.047 | / | neap | 6:28 |  |
| | | 8.489
55.062 | / | | | |
| MTC | 23/05/2018 | 8.461
55.039 | / | neap+1 day | 6:19 |  |
| | | 8.474
55.035 | / | | | |
| | | 8.499
55.046 | / | | | |
| | 24/05/2018 | 8.505
55.056 | / | neap+2 days | 5:56 |  |
| ITI | 29/05/2018 | 8.464
55.047 | / | spring | 7:11 |  |
| | | 8.489
55.062 | / | | | |
| OTI | 30/05/2018 | 8.418
55.059 | / | spring+1
day | 12:24 |  |
| | | 8.428
55.071 | / | | | |

The main wind direction during the cruises was around the 90° (East) sextant, with the most frequent intensities in the range of 5 to 10 m/s. During 24[th] of May the wind was blowing strongly from the east nearly during the whole cruise. Exceptions were the cruise on May 23 concerning the direction, when were observed winds from the NNW (330°), and the cruise on May 29 with mostly frequent intensity ranging between 10 to 15 m/s.

Table 2 in the manuscript shows that for all solutions except FES, the correlation coefficients are higher during spring tides as well as in the deepest part of the domain despite the quite strong winds often ranging from 10 to 15 m/s 29th and 30th of May. The largest velocity errors for all solutions occur when tidal velocities are small as well as during the tidal state change, so we can guess that at these moments wind, wave and baroclinicity effect become much more pronounced. On May 23 and May 24, the measurements were performed on nearly the same side; but the May 24 correlation coefficient for the 'v' component is relatively small. In this case we can really say that the reason is permanent wind from the east. However in this zone, tides seem to be explained more than 80 % (or 90 % or more in case of a spring tide) of the dynamics in case of absent storm (more than 20 m/s) and blowing continuously in one direction winds.

We believe that the wind forcing will add approximately the same contribution to error for all boundary conditions used based on very high correlation coefficients.

The FFT analysis of the modeling results was done after the stabilization of the total energy behavior. The elevation signal from TG was analyzed based on the whole year time-series. Due to the fact that the frequencies induced by wind are very high, we believe that this is high quality analysis. However, of course, some errors are there, which are hard to correct.

The 'wind' table has been added to the Appendix.

line 195: Is Cd constant in space? If so, is that a realistic configuration for the bight?

Tide gauge comparisons: It would be useful to have example time series comparing

the observations and the model to give a better idea of the model's performance. A

Taylor diagram could also be used.

115

Yes, we  took Cd as a constant for the region due to the fact that the seabed habitat map is not ready yet for the area, there are nearly no observations in the shallow part of the region  and considered area is relatively small. Therefore we think that such a decision is justified.

The reconstructed tidal elevation from observations based on FFT and elevation based on model runs
120     visually are nearly identical. The intercomparison example of the observed and  modeled amplitudes at the gauging station + Taylor diagrams for amplitude and phase are given below.  The diagrams were added to supplementary materials.

[Figure]

125

Fig. Demonstration of the FFT analysis results at List TG.

[Figure]

Taylor diagrams based on observed and modelled data at 3 gauging stations, '1' and '2' indicate simulation results on curvilinear and unstructured grids respectively: a) for amplitude; b) for phase.

130

line 219: Do you use a sponge layer? If so, please describe it in Section 2.
Thank you for the comment.

To avoid errors due to the inconsistency between the character of equations and the specified open

135 boundary conditions (prescription of tidal elevation only), a 3-km sponge layer has been introduced. It gradually turns off the advection of momentum and viscosity in the vicinity of the open boundary (Androsov et al., 1995). The appropriate information is added to the manuscript.

Androsov, A.A., Klevanny, K.A., Salusti, E.S., Voltzinger, N.E.: Open boundary conditions for horizontal 2-D curvilinear-grid long-wave dynamics of a strait, Advances in Water Resources, 18, 267–276, 10.1016/0309-1708(95)00017-D, 1995.

140

Baroclinicity: The tidal dynamics, e.g. tidal ellipses, are probably affected by density
and stratification effects which in the present study is neglected. Can you argue that
baroclinic effects are negligible in this system?

145 The water column in the domain of interests is generally well mixed, weak strain induced periodic stratification only occurs at the end of the flood in some subareas (Villarreal et al., 2005; Simpson et al., 1990; Purkiani et al., 2015). So we think that for the **depth-averaged** ellipse maps we can omit baroclinic effect. The presence of the weak density gradient in the system has an influence to the local

circulation, especially in the Lister deep zone. But there the tidal residual circulation is large and would

150 dominant the dynamics. We think that our results are valid in case of baroclinic effects are turned on, however  further consideration of the SPM budget would be not possible without considering baroclinicity (e.g., Burchard et al., 2008).

Burchard, H., Flöser, G., Staneva, J. V., Riethmüller, R. and Badewien, T.: Impact of density gradients on net sediment

155 transport into the Wadden Sea, J. Phys. Oceanogr., 38, 566 – 587, https://doi.org/10.1175/2007JPO3796.1, 2008.
Purkiani, K., Becherer, J., Flöser, G., Gräwe, U., Mohrholz, V., Schuttelaars, H. M. and Burchard, H.: Numerical analysis of stratification and destratification processes in a tidally energetic inlet with an ebb tidal delta, J. Geophys. Res-Oceans, 120, 225– 243, https://doi.org/10.1002/2014JC010325, 2015.
Simpson, J.H., Brown, J., Matthews, J. et al. Estuaries (1990) 13: 125. https://doi.org/10.2307/1351581.

160 Villarreal, M.R., K. Bolding, Burchard, H., and E. Demirov, 2005. Coupling of the GOTM turbulence module to some three-dimensional ocean models, Marine Turbulence: Theories, Observations and Models, Baumert, H. Z., J. H. Simpson, and J. Sündermann, Eds., Cam-bridge University Press, Cambridge, 225–237.

165

Section 5.2: Larger dissipation in the unstructured grid could also be due to better resolved intertidal dynamics that are inherently dissipative. Presumably also the bathymetric features are quite different in these two grids.

170 Thank you for the comments. We agree that better resolved intertidal dynamics can be an additional factor of the larger dissipation. This part of the dissipation can be largely traced in the behavior of the bottom friction and energy change adds of the balance. The conclusion about larger dissipation on the unstructured grid is made based on analysis of the energy imbalance. We should stress that the unstructured grid is more dissipative numerically anyway with our type of discretization, this is well

175 proofed result (e.g. Danilov and Androsov, 2015; Androsov et al., 2019).

The additional comments have been added to the text.

Danilov, S. and Androsov, A.: Cell-vertex discretization of shallow water equations on mixed unstructured meshes, Ocean Dynam., 65, 33 – 47, https://doi.org/10.1007/s10236-014-0790-x, 2015.

180 Androsov, A., Fofonova, V., Kuznetsov, I., Danilov, S., Rakowsky, N., Harig, S., Brix, H., and Wiltshire, K. H.: FESOM-C v.2: coastal dynamics on hybrid unstructured meshes, Geosci. Model Dev., 12, 1009 – 1028, https://doi.org/10.5194/gmd-12-1009-2019.

The authors conclude that the model results "converge" to a realistic solution (line 81,

 478, abstract), based on the presented simulations with two different grids. The authors also conclude that the curvilinear grid has lower dissipation making it thus better suited for baroclinic studies. I find these conclusions somewhat premature: Only two different grids were used, which at the same time had different element types (triangles and quads), resolution and topology (unstructured and curvilinear), as well as (I presume)

190 bathymetry. As such, it is really quite hard to infer what grid properties cause the observed change in model performance. The grid sensitivity study should be extended to better address the effects.

We made a study  of convergence of the solution on meshes of different configurations (quadrilateral,
195 triangular and mixed) for the studied region in the work of Androsov et al., 2019.  The study is based on coarser meshes and bathymetry data  and with only M2 forcing. However, it  provides a comparative analysis of energy characteristics as well as a histogram of errors of dynamical characteristics on meshes of different configurations. As was mentioned above it is proofed fact that the quadrilateral (curvilinear) has lower numerical dissipation with our desritization. We agree that this a premature to
200 talk about which grid will better in case of baroclinic study, it is actually should be checked in further work.  We have decided to write additional small article dedicated to the reproduction of the nonlinear effects on the grids of different structure. The bathymetry matrix was the same for both grids. We have added the reference to the manuscript and additional notes.

205

Androsov, A., Fofonova, V., Kuznetsov, I., Danilov, S., Rakowsky, N., Harig, S., Brix, H., and Wiltshire, K. H.: FESOM-C v.2: coastal dynamics on hybrid unstructured meshes, Geosci. Model Dev., 12, 1009 – 1028, https://doi.org/10.5194/gmd-12-1009-2019.
Technical corrections
210 line 123: 1/30 *degree*
line 182: this paragraph is duplicate of the previous one.
Figure 3 a: The ratio being thus defined it would be more appropriate to call it "weight
of linearity" instead. The unit (m) is wrong.
Table 3: Add units. What do "RMSD", "amp(cm)", "ph(_)" stand for?
215 Figure 9: Add units.

Thank you a lot! Done.

**Referee 2**

The article is a detailed study of the nonlinear dynamics of the tide in the shallow bay of the North Sea, which is characterized by a significant area of the intertidal zone. As a research tool, a relatively new FESOM-c model is used, which approximates the governing equations by the finite volume method and is able to work on hybrid unstructured computational grids. New data on the bathymetry of the bay and tidal currents are also presented. The authors of the study set a rather difficult goal, analyzing the results of observations and modeling for the summary tide instead of effects of nonlinearity for the single harmonic tide. In the latter case, it would be easier to estimate the contribution of the main mechanisms of nonlinearity (shallow depth, advection, and quadratic friction) on the structure of tidal asymmetries. Nevertheless, the results presented in the article are of real interest, since they provide a serious basis for analyzing the features of sediment dynamics and the formation of a variable bottom relief. I would recommend this paper to be published after a little revision. I hope the comments below help improve the manuscript.

Dear Referee 2,

Thank you so much for the very valuable comments! Please, find the answers below. We agree, that further study should be dedicated to the analysis of the separate mechanisms of nonlinearity in the domain based on grids with different structure.

Kind regards, Vera on behalf of the co-authors

Specific comments:

Line 100: Are only 10 sigma vertical layers enough for 3D simulations? In other words, is the numerical solution dependent on the number of vertical layers?

Thank you for the question. The water column in the domain of interests is generally well mixed, weak strain induced periodic stratification only occurs at the end of the flood in some subareas (Villarreal et

al., 2005; Simpson et al., 1990; Purkiani et al., 2015). Based on available observations we can conclude that the vertical structure of the velocities and turbulent characteristics are relatively simple (Burchard et al., 2008, Purkiani et al., 2015)

250

As soon as we have concentrated our attention mainly on the depth-averaged dynamics and near-bottom dynamics (our vertical layers are crowded near the bottom) and the area of consideration is relatively shallow, we have agreed on 10 levels.

255  Burchard, H., Flöser, G., Staneva, J. V., Riethmüller, R. and Badewien, T.: Impact of density gradients on net sediment transport into the Wadden Sea, J. Phys. Oceanogr., 38, 566 – 587, https://doi.org/10.1175/2007JPO3796.1, 2008.
Purkiani, K., Becherer, J., Flöser, G., Gräwe, U., Mohrholz, V., Schuttelaars, H. M. and Burchard, H.: Numerical analysis of stratification and destratification processes in a tidally energetic inlet with an ebb tidal delta, J. Geophys. Res-Oceans, 120, 225– 243, https://doi.org/10.1002/2014JC010325, 2015.
260  Simpson, J.H., Brown, J., Matthews, J. et al. Estuaries (1990) 13: 125. https://doi.org/10.2307/1351581.
*Villarreal*, *M.R.*, *K. Bolding*, *Burchard*, *H*., and *E. Demirov*, 2005. *Coupling* of the *GOTM turbulence module* to *some three-dimensional ocean models*, Marine Turbulence: Theories, Observations and Models, Baumert, H. Z., J. H. Simpson, and J. Sündermann, Eds., Cam-bridge University Press, Cambridge, 225–237.

265  Section 2.3 (Open boundary conditions) Please specify the period of model calculations (dates).

The spin-up period for all simulations was three months with a criteria of the stabilization of the energy behavior. Due to the fact that paper considers only the tidal dynamics for the analysis we took last two

270  full tidal periods - 59 days. We simulated the tidal dynamics in 2018, which is expressed in Doodsen correction of the prescribed amplitudes and phases, therefore we were able to compare observed and modeled velocities second to second for end of May 2018.

The additional setup details have been added to the manuscript.

275  Line 165 (Data. 3.2 Tide gauge (TG) data): VidaTG station is located in the intertidal zone, and so during low tide ( when the seabed is exposed) the continuous (quasi harmonic) time series is greatly distorted. (see for example, https://www.emodnetphysics. eu/map/platinfo/piroosplot.aspx?platformid=9015&60days=true). In fact, this is time series with data gaps. Nevertheless, the authors used classical harmonic analysis

280  for this station in the validation of the model. Did authors take into account the

peculiarity of the tide in this station when analyzing the results? (see also the comment

below on Section 4.5.2).

Thank you for the question. VidaTG station is situated in the intertidal zone, however itself is situated in
285   the quite deep channel, therefore there is no distortion of the time series (Fig. 1).

[Figure]

Figure 1.  The elevation time series at the Vidaa TG and results of the classical harmonic analysis.

290   Table 2. The results of the inter comparison show rather large RMSD values (compared

with the velocity values themselves). Apparently, this is due to the neglect of wind induced

fluctuations in the simulation. The question arises: Is inter comparison in

Table_2 appropriate?

295   Thank you for the question. We think that the wind forcing will add approximately the same
      contribution to the error for all boundary conditions used based on very high correlation coefficients. In
      considered zone tides seem to be explained more than 80 % (or 90 % or more in case of a spring tide) of
      the dynamics in case of absent storm (more than 20 m/s) and blowing continuously in one direction
      winds. On this basis, we give this comparison in Table 2, the purpose of which is to select those
300   boundary conditions that have the smallest absolute error compared to the observational data.

Line 330: It seems to me that the use of the term "seiche" is not entirely appropriate in

this context, since we are dealing with forced fluctuations. It would be more correct to speak about oscillations as near standing wave.

305

We replaced the term "seiche" with a more general term – "standing wave".

Line 350: It is desirable to immediately emphasize that the results of Fig. 8 relate to the sea level (not currents).

310    Thank you for the comments. We have put additional notes to the figure caption.

Line 359: Frankly, I do not see indication on degenerate amphidromic point in external area. Yes, there is a slight closeness of phase contours lines (caused by proximity to the strait and refraction of the tidal wave), but there is no decrease in the tide amplitude as it is usually in nodal (amphidromic) zone. The effect of the capture of a Poincare

315    wave is interesting, but requires explanation or reference.

Thank you. Absolutely correct remark, this picture does not clearly show the presence of amphidromic point, because we do not see a concentric decrease in the level. We have excluded this conclusion from the text.

320

Line 365-370: It is interesting how the authors distinguish between the different duration of the ebb and tide caused by the nonlinearity from the effect of the sum of harmonics of various periods. The last effect is called the Diurnal Inequality of tide, and it is not connected with non-linearity.

325

Thank you very much for the very useful remarks! The features represented in Fig. 9 (level panel) show the mean pattern for the full tidal cycle (29.5 days). However, we should definitely pointed out that the reason of the variations in Fig. 9 (right panel) is not only non-linear effects, but also Diurnal Inequality of tide. The additional comments have been added to the text.

330

Line 405-410: To explain the effect of the dominance of tide velocities over ebb ones,

it is not necessary to use phase speed inequality (SQRT (gh)). There is a simple explanation

for this effect: the bottom friction dampens the flow more efficiently in smaller

depth (during low ebb).

335

We agree to the comments. However, in this zone, the depth is relatively large and residual circulation is smallest compared to the surrounding area.

The mentioned effect induced by bottom friction can be traced in the zone 2, where we have an extensive intertidal zone and maximum amplitudes of the semidiurnal tides compared to other zones,

340    here the flood dominance can be explained by the major role of non-linear friction effects and non-linearity in changes of the water-layer thickness.

Section 4.5.2 (Line 412). Again about the accuracy of harmonic analysis in the intertidal

345    zone. Indeed, in this zone, at low tide, the bottom is exposed, and this means that data
gaps appear in the model time series. In this case, the classical harmonic analysis can
give inaccurate results for the amplitudes and phases of the waves. It seems that it
would be more logical to exclude intertidal zones from the analysis of results (at least
in terms of the results of harmonic analysis). How was this fact taken into account?

350    Otherwise it's necessary apply special procedures to avoid mistakes.

Thank you a lot for the comment. We agree with the remark. We have decided to consider intertidal zone, because it occupies at about 40 % of the whole considered area. We performed our simulations with time step of about 1 s, we did not do any averaging and used output every couple of seconds to

355    perform the analysis. Therefore, analyzing last two full tidal cycles (29.5 × 2 days), we had large time-series. Also we have performed sensitivity test (modifying the analyzed period), which showed that the results of the analysis are justified.

Section 6 (Summary). Unfortunately, the question of the sensitivity of the numerical simulation to the accuracy of bathymetric data remained outside the discussion. In particular, how important is the effect of new bathymetric data in the strait on simulation results. A comparison of the solutions with the old and new bathymetry would answer this question, and perhaps provide a serious justification for the need for new good bathymetry for the whole area.

Important note. Thanks. At a preliminary stage, comparison analysis of results of the computation for three bathymetric databases is carried out: 200 m (H. Burchard, personal communication), 100 m and with a resolution of 50 m (L. Sander, personal communication). The meshes used in this analysis have a rectangular structure and the spatial resolution corresponding to the bathymetry info. Note, that 200 m bathymetry is largely smoothed compared to 50 m bathymetry product. Results were analyzed on fields of residual circulation in a tidal cycle of M2 wave. The residual circulation based on high quality bathymetry data considerably differs from one based on a coarser bathymetry (figures below). Vortex structures have some space shift, a considerable intensification on detailed bathymetry in comparison with smoothed one (figures below).

[Figure]

Bathymetry. Left – 200 m resolution; middle – 100 m; right – 50 m.

380

[Figure]

[Figure]

[Figure]

Residual circulation near bottom. M2 wave. Upper panel – 200 m bathymetry resolution; middle panel –
100 m resolution; bottom panel – 50 m.

Technical corrections:

Lines 175 and 182: Two paragraphs contain the same information.
Line 245: 29.5 days is rather a lunar (synodic) month than a tidal cycle
Line 335: Direction of rotation (not orientation) of the tidal ellipse is determined by the
sign of the ellipticity. (Orientation is rather the inclination of its main axis of the ellipse).
Line 372: Replace "(level panel)" by "(left panel)"
Figure 5. Please explain what the following sentence means: "The length of vectors on
the maps is scaled based on the square root method."
Figure 8 .: "The dark blue color indicates zone, where topography features are above
sea level." ....... above the highest sea level ???

Thank you a lot! Done

**Referee 3**

Dear Referee 3,

Thank you for the comments. Please, find the answers below.

405

Kind regards, Vera on behalf of the co-authors

This is a high resolution hydrodynamic modelling study of a region of the Wadden Sea. A new bathymetric product for the region has been incorporated as model input, allowing for higher resolution

410 simulations to be conducted – but what are the implications of this to the wider field? Overall, I felt that although the methods were sound, the paper lacked novelty and read as more of a 'site characterisation' study than a piece of novel research. My main comment is that the authors could do more to convince the readers that this is an interesting region to study, and to describe more clearly what this paper adds to the field of knowledge on the area/ implications for future modelling studies. Further still, there are

415 numerous ambiguous and subjective statements made. Because of the lack of explicit motivation for the importance/relevance of this work, I found the lengthy results section difficult to follow. On this basis, I would recommend that the manuscript be returned to the authors for a major (editorial) revision, to include the re-working of many of the figures. My apologies that it has taken me a long time to return this review.

420 **Importance/relevance**

The tidal residual circulation and asymmetric tidal cycles largely define the transport and accumulation of sediment and the distribution of bedforms in the bight. Largely due to gap in understanding of the asymmetric tidal cycles phenomena, in the literature you can find different values of suspended matter fluxes and the sediment budget in/of the bight (e.g., Boldreel et al., 2010; Burchard et al., 2008; Hayes,

425 1980; Nortier, 2004; Postma, 1967). Even the estimate of the relatively simple value as the mean water volume entering the basin during flood and leaving during ebb in a mean sense varies from 4 to 6.3 $*10^8$ m$^3$ (Bolaños-Sanchez et al., 2005; Gätje and Reise, 1998; Gräwe et al. 2016; Lumborg and Windelin, 2003; Nortier, 2004; Pejrup et al., 1997; Purkiani et al. 2015).

**Novelty, major points**

scientific

- Four zones with fundamentally different asymmetry structures were indicated during tidal asymmetry analysis.
- Tidal asymmetry maps were firstly introduced and constructed.
- Tidal ellipses, energy fluxes and residual circulation maps were constructed and analyzed.

Based on this analysis you can largely judge about the velocity behavior, water parcel trajectories and bedform peculiarities in the area.

methodical

- For the first time the bathymetry product with such a high resolution was used for the area.
- The two new grids with different structure were constructed and convergence of the results were shown.
- The best tidal solution at the open boundary for the area was identified.

Comment to the hand-written remark:

- Spaces must be included between number and unit (e.g. 1 %, 1 m) according to the journal rules.
- Please, consider that the quality of the pictures in the manuscript is not the best to reduce the size of the pdf.
- The manuscript was proofread by the native speaker except the Discussion Section.

Bolaños-Sanchez, R., Riethmüller, R., Gayer, G. and Amos C. L.: Sediment transport in a tidal lagoon subject to varying winds evaluated with a coupled current-wave model, J. Coastal Res., 21, e11 – e26, https://doi.org/10.2112/03-0048.1, 2005.

Boldreel, L. O., Kuijpers, A., Madsen, E. B., Hass, H. C., Lindhorst, S., Rasmussen, R., Nielsen, M.G, Bartholdy, J. and Pedersen, J. B. T.: Postglacial sedimentary regime around northern Sylt, South-eastern North Sea, based on shallow seismic profiles. Bull. Geol. Soc. Den., 58, 15 – 27, 2010.

Burchard, H., Flöser, G., Staneva, J. V., Riethmüller, R. and Badewien, T.: Impact of density gradients on net sediment transport into the Wadden Sea, J. Phys. Oceanogr., 38, 566 – 587, https://doi.org/10.1175/2007JPO3796.1, 2008.

Gätje, C. and Reise, K. (Eds.): Ökosystem Wattenmeer: Austausch-, Transport- und Stoffumwandlungsprozesse, Springer, Berlin, Heidelberg, Germany, https://doi.org/10.1007/978-3-642-58751-1, 1998.

Hayes, M. O.: General morphology and sediment patterns in tidal inlets, Sediment. Geol., 26, 139 – 156, https://doi.org/10.1016/0037-0738(80)90009-3, 1980.

Gräwe, U., Flöser, G., Gerkema, T., Duran-Matute, M., Badewien, T. H., Schulz, E. and Burchard, H.: A numerical model for the entire Wadden Sea: Skill assessment and analysis of hydrodynamics, J. Geophys. Res-Oceans, 121, 5231 – 5251, https://doi.org/10.1002/2016JC011655, 2016.

Nortier, R. J.; Morphodynamics of the Lister Tief tidal basin. M. S. thesis, TU Delft, Netherlands, pp. 76, available at:
465    http://resolver.tudelft.nl/uuid:89da7250-9acb-429d-bb71-d7dbc60e7755, 2004.

Pejrup, M., Larsen, M. and Edelvang, K.: A fine-grained sediment budget for the Sylt-Rømø tidal basin, Helgolander Meeresun., 51, 253 – 268, https://doi.org/10.1007/BF02908714, 1997.

Postma, H.: Sediment transport and sedimentation in the estuarine environment, in: Estuaries, edited by: Lauff, G. H., American Association for the Advancement of Science, Washington, DC, 158 – 179, 1967.

470    Purkiani, K., Becherer, J., Flöser, G., Gräwe, U., Mohrholz, V., Schuttelaars, H. M. and Burchard, H.: Numerical analysis of stratification and destratification processes in a tidally energetic inlet with an ebb tidal delta, J. Geophys. Res-Oceans, 120, 225– 243, https://doi.org/10.1002/2014JC010325, 2015.

Lumborg, U. and Windelin, A.: Hydrography and cohesive sediment modelling: Application to the Rømø Dyb tidal area, J. Marine Syst., 38, 287 – 303, https://doi.org/10.1016/S0924-7963(02)00247-6, 2003.

475

I have included some general comments below. I have made many hand-written notes on the manuscript that should point to where I saw the above issues manifested (please see the attached scanned document). I stopped correcting the language on page 8, but ambiguous terms remain which need to be edited out. What are "patchy dynamics"? What are velocities which are "nearly always larger"? Please
480    quantify, expand, elaborate, justify and support your statements.

Thank you for the comments. We have included the corrections to the manuscript based on your notes.

General comments

1. The abstract could do with opening with a broader, overarching introduction of the topic, rather than launching into what is done here.

485    The abstract was modified.

2. The language used was subjective in places (e.g. " the quite good results") – please take care to make all statements objective.

We made the corrections. We left some 'subjective' sentences, where we have directly  link to the numbers (such as a reference to the tables).

3. Can you avoid referring to the grids as first and second grids (which is confusing), but instead refer to them as 'curvilinear' and 'unstructured' (or similar, as appropriate).

Done

4. line 194 – what were these additional experiments? It's fine not to show them, but what were they and how did you get to the conclusion that you draw?

Thank you for the comment, now we can add a reference, where these experiments are described. It has been done.

5. Line 200 – same or worse results? Please elaborate.

Here, the major point that we were not able to improve the correlation coefficient and RMSD varying further bottom friction coefficient. The changes in the text have been done.

6. Line 224-225, please can we have a range of depth layer thicknesses and explain how these were parameterized, e.g. terrain-following?

The 10-sigma (terrain-following) vertical layers were prescribed. The distribution of the sigma layers satisfies the parabolic function. The minimum possible thickness of the vertical layer is ~1 cm.

sigma(k)=-[(k-1)/(kb-1)]^2, kb is a total amount of sigma layers, k is current layer (k=1..kb), sigma=-1 at the bottom, sigma=0 at the surface.

7. Line 237 – " the same probably holds". This is ambiguous, and does not fill me with confidence in the rest of your interpretations, please can you clarify, and back this statement up?

We agree that this statement, probably, needs a detailed interpretation, so we have decided to remove it.

8. Please revise the figure colour scales! Please improve upon the multi-coloured ones, as a matter of priority. Figures 3, 6, 8 and 9, in particular, are impossible to interpret. 9. Figure 4, pink and red difficult to differentiate, please consider dashed lines or something, so that these are meaningful even in black and white. 10. The vector plots cannot be interpreted, these need to be made clearer. Fewer arrows perhaps? Please reconsider how these data are presented.

We changed the figures based on your comments and hand-written notes.

11. The discussion is short and lacking depth and detail. What have you found, what are the implications, and for whom? Please revise and improve.

The discussion of the manuscript was modified. We did not go into the details about each considered topics, because they are too large and require additional detailed study.

[revised manuscript text omitted]